# Dual-granularity Dense Reward over Generated Episodic Guidance for Embodied Reinforcement Learning

## Abstract

Designing suitable rewards poses a big challenge in reinforcement learning (RL), especially for embodied manipulation. Trajectory success rewards are suitable for human judges or model fitting, but the sparsity leads to low RL sample efficiency. While recent methods have effectively improved RL via dense reward, they rely heavily on high-quality human-annotated data or abundant expert supervision. For issues above, this paper proposes Dual-granularity self-supervised reward via generated Episodic Guidance (DEG), a novel framework to seek sample-efficient dense rewards without requiring human annotations or extensive supervision. Leveraging the prior knowledge of large video generation models, DEG only needs a small amount of expert videos for domain adaptation to generate dedicated task guidance for each RL episode. Then, the proposed dual-granularity reward that balances coarse-grained exploration and fine-grained matching, will guide the agent to efficiently approximate the generated guidance video sequentially in the self-supervised latent space, and finally complete the target task. Extensive experiments on 18 diverse tasks across both simulation and real-world settings show that DEG can not only serve as an efficient exploration stimulation to help the agent quickly discover sparse success rewards, but also guide effective RL and stable policy convergence independently.

## 1. Introduction

Although supervised imitation learning (behavior cloning) has shown considerable policy-shaping capabilities in robotics control (Chi et al., 2025; Chen et al., 2024a), autonomous driving (Kiran et al., 2021; Jiang et al., 2025), and

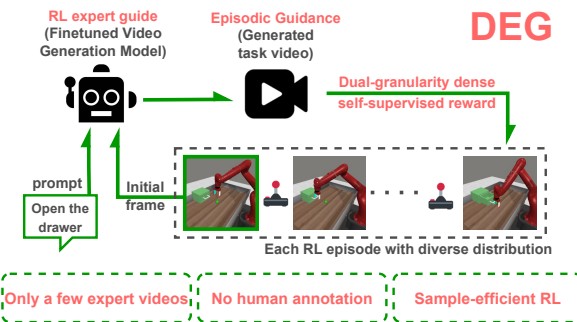

*Figure 1.* The pipeline of the proposed DEG. Without requiring human annotations or extensive supervision, DEG enables sample-efficient RL via dual-granularity self-supervised dense reward based on the generated episodic video guidance.

the currently booming field of large language model fine-tuning (Hu et al., 2022; Li et al., 2025), two factors severely limit its upper limit of policy performance and application scenarios: (i) its reliance on large amounts of expert data, and (ii) its susceptibility to compounding errors (Zare et al., 2024). In contrast, RL can effectively address these two issues, making it a focus once again after game artificial intelligence (AI) (Mnih et al., 2015). Despite its advantages, RL also faces a tricky problem: it requires sufficient environmental interactions to support the agent's trial-and-error and policy refinement. To solve this problem, researchers have made numerous attempts in different directions, including improving representation modules (Yarats et al., 2022; Laskin et al., 2020), expanding training data (Seo et al., 2023; Deng et al., 2022; Hafner et al., 2025), exploring multi-task potential (Liu et al., 2025b; Sun et al., 2025), and so on. These progress demonstrate that, driven by well-designed rewards, RL is able to support sample-efficient policy learning processes in various scenarios.

However, how to design appropriate rewards for RL poses another challenge. The most primitive approach is to assign binary success-or-failure rewards to entire trajectories (Luo et al., 2025a). Such rewards are not only easy for humans to judge or provide feedback on, but also simple for (large) reward models to learn and fit (Lee et al., 2026). Yet their sparsity often leads to low sampling efficiency in RL, especially in domains with large state and action spaces. Therefore, suitable dense rewards are often the

[1]Anonymous Institution, Anonymous City, Anonymous Region, Anonymous Country. Correspondence to: Anonymous Author <anon.email@domain.com>.

key to achieving sample-efficient RL. Currently, common public benchmarks (Tassa et al., 2018; Mnih et al., 2015) in the RL field usually provide effective dense feedback, but this is often the result of painstaking manual design by human experts (Cobbe et al., 2020). In manipulation tasks with stage-varying goals and complex interactions (Yu et al., 2020), manually designing dense rewards is even quite challenging for experts (McLean et al., 2025). Thus, some recent progress has aimed to realize automated dense reward design for manipulation tasks, attempting to derive dense rewards from expert demonstrations by leveraging advanced techniques such as adversarial learning (Karnan et al., 2022), generative AI (Chen et al., 2025a), and vision-language large models (Tan et al., 2025). Nevertheless, these methods either rely on large volumes of expert supervisions, or require massive amounts of manually annotated data.

The above discussion naturally leads to a question: *can we design dense rewards for sample-efficient RL with minimal use of expert data and no reliance on manual annotations?* A novel framework named Dual-granularity self-supervised reward via generated Episodic Guidance (DEG) is proposed, trying to answer this open question. DEG leverages the prior knowledge of large models to overcome the challenge of scarce data. It enables open-source video generation models to accomplish domain adaptation with a tiny amount of expert videos (only 3-5 clips) through fine-tuning, familiarizing the generator with the dynamics and workflow of target tasks. The fine-tuned video generation model then acts as an expert guide, producing an expert guidance video tailored to each episode with changing initial states in the RL process. Via self-supervised learning, DEG aligns the noisy generated videos with high-fidelity real observations in the semantic latent space, and calculate a novel dual-granularity reward that comprises a coarse-grained exploration incentive and a fine-grained matching incentive. The former drives the agent to roughly imitate the guidance video sequentially, which is achieved via a relatively loose threshold judging imitation progress. The latter, by contrast, focuses on rewarding precise imitations that closely match the guidance video within the rough movements. It ensures the attention on fine-grained details (such as delicate interactions with objects), leading to refinement of the policy from rough to accurate. Through the organic integration of these two components, the dual-granularity reward guides the agent to efficiently and accurately approximate the generated guidance and ultimately complete the target task, as shown in Figure 1.

We conduct extensive experiments on 18 manipulation tasks across both simulation and real-world settings. DEG can serve as an efficient intrinsic stimulation to help the agent quickly discover sparse success rewards, matching (or even outperforming) manually designed dense rewards by human experts. It can also guide effective RL and achieve

stable policy convergence independently, surpassing state-of-the-art reward engineering approaches with much less supervision required. We summarize the contribution of our paper as follows:

- We propose a novel framework, dubbed Dual-granularity reward via generated Episodic Guidance (DEG), for RL dense reward design. With minimal use of expert videos and no reliance on manual annotations, DEG reward can not only help the agent quickly discover sparse success rewards, but also independently guide effective RL and achieve stable policy convergence.

- Through finetuning the open-source video generation models, DEG requires only a few expert videos to generate a dedicated task guidance for each RL episode with diverse initial states. A novel dual-granularity self-supervised reward function that balances coarse-grained exploration and fine-grained matching, will drive the agent to efficiently imitate the guidance and finally complete the target task.

- We conduct extensive experiments on a set of challenging manipulation tasks, across both simulation and real-world settings. With the aid of sparse success reward, DEG achieves efficient RL across all 18 tasks, matching the performance of dense rewards handcrafted by human experts. In addition, DEG also outperforms state-of-the-art baselines on twelve reward-free tasks, with much less expert supervisions required.

## 2. Related Works

### 2.1. Manipulation Policy Learning

Robotic manipulation has seen significant progress through behavior cloning, with recent architectures like Diffusion Policy (Chi et al., 2025) and ACT (Zhao et al., 2023) demonstrating high-performance imitation in modeling complex behavior distributions from expert data. However, BC remains constrained by the quality of demonstrations and struggles with sub-optimal data (Schulman et al., 2017; Kostrikov et al., 2022). To enable self-improvement, RL is often employed for policy fine-tuning (Schulman et al., 2017; Luo et al., 2025b; Kostrikov et al., 2022). By interacting with the environment and receiving reward signals, agents can refine their behavior beyond the expert demonstrations. Despite its potential, RL manipulation tasks are frequently hindered by the "reward engineering" bottleneck, where defining reward signals in unstructured settings requires manual effort or privileged state information that is not practical (Chen et al., 2025b; Singh et al., 2019). To overcome this, our approach focuses on synthesizing reward signals for manipulation policy improvement.

## 2.2. Learning from Videos

Based on the relationship with the task domain, videos can be divided into cross-domain videos and in-domain videos. Cross-domain videos are usually abundant in quantity, but there is a gap between them and the target task in terms of observation and action spaces (Liu et al., 2025b). Therefore, some existing works often use them as an auxiliary to improve the performance or efficiency of existing policy learning pipelines, such as enhancing cross-domain representation (Xiao et al., 2022) or pre-training universal latent representations (Ye et al., 2025; Chen et al., 2024b). In contrast, in-domain videos have an extremely high density of available information, but they are difficult to obtain directly from the Internet and need to be manually acquired at a high cost. Although many works have proved that expert videos can serve as the main (Schmidt & Jiang, 2024; Ye et al., 2022) or even the only source of supervision signals (Liu et al., 2025c; Giammarino et al., 2024; Chen et al., 2025a) to directly support the policy convergence on target tasks, they often rely on the assumption of sufficient video data and ignore the key metric of video data efficiency (Liu et al., 2023; 2025a). With the rapid development of large models, it has become possible to perform large-scale pre-training on cross-domain videos to generate in-domain videos (Wan et al., 2025; Seawead et al., 2025; Wu et al., 2025). Recent progress (Jang et al., 2025) clearly demonstrates the feasibility of using video generation models as data generators to enhance video behavior cloning. Inspired by this, GEG first proposes to use a large video generator as an episode guide and provides corresponding rewards during the RL process.

### 2.3. RL Reward Design

Exploration is crucial in RL, which motivates researchers to investigate intrinsic rewards for enhancing exploration (Burda et al., 2019). These rewards are generally decoupled from specific tasks while proportional to the agent's exploration of the state space (Liu & Abbeel, 2021). They can not only serve as plug-and-play modules to boost the RL early-stage exploration (Yarats et al., 2021; Liu et al., 2025b) but also collect unsupervised pre-training data for multi-task RL (Sun et al., 2025). However, the task-agnostic nature results in their limited ability to handle complex state spaces and solve complex long-horizon tasks (Chen et al., 2025a), which has spurred numerous works to focus on task-specific reward design. Inverse RL methods (Torabi et al., 2018; Giammarino et al., 2025) automatically infer expert rewards from expert demonstrations. They enable high-quality imitation of expert policies without additional human annotations while requiring abundant expert data and extensive environmental interactions (Zare et al., 2024). Large reward models have achieved remarkable performance recently in both sparse feedback (Lee et al., 2026; Sontakke et al., 2023; Zhang et al., 2025) and dense incentive design (Tan et al.,

2025; Zhai et al., 2025), but they either suffer from reward hacking or rely on massive human annotations and training resources. In contrast, DEG leverages video generation models with prior knowledge as RL guides to realize automated and accurate reward design, while breaking free from the reliance on large-scale expert data and human annotations.

## 3. Methodology

The goal of this paper is to derive the task-specific dense reward function for an environment with only sparse reward or even no reward. The former environment can be represented as a sparse-reward Markov Decision Process (MDP) (Bellman, 1957) $\mathcal{M} = (\mathcal{O}, \mathcal{A}, \mathcal{P}, \mathcal{R}^s, \gamma, d_0)$, where $\mathcal{O}$ denotes the visual observation space, $\mathcal{A}$ denotes the action space, $\mathcal{P}$ denotes the transition function, $\mathcal{R}^s$ denotes the sparse success reward function, $\gamma$ is the discount, and $d_0$ is the distribution of the initial observation. The latter can be correspondingly expressed as a reward-free MDP $\mathcal{M} = (\mathcal{O}, \mathcal{A}, \mathcal{P}, \gamma, d_0)$ where the symbols share the same meanings as those in the former. With the help of a well-pretrained video generation model $G$, DEG only requires a few expert videos $\mathcal{V}$ for reward calculation.

DEG first obtains a robust task-specific guide $G'$ based on a well pre-trained video generation model $G$ and a small amount of expert videos $\mathcal{V}$ (**Section 3.1**). The guide $G'$ will generate unique task guidance videos for each episode with distinct initial positions in the following RL process. Considering that noise is inevitably present during video generation, DEG aligns the generated and real image observations in the latent space via self-supervised learning (**Section 3.2**), and calculates a novel dual-granularity dense reward. This reward balances coarse-grained exploration and fine-grained matching (**Section 3.3**), driving the agent to efficiently and accurately follow the instructions provided by the guide to complete the target task.

### 3.1. Training Task Guide

We leverage the extensive prior knowledge of pre-trained large models to mitigate the problem of incomplete expert data distribution. Specifically, we select Wan2.1-I2V-14B (Wan et al., 2025), an advanced and open-source image-to-video model, as our backbone. We find that without finetuning, it already possesses the ability to understand the semantics of environments and objects, while lacking knowledge on the specific dynamics relevant to the target task (e.g., the degrees of freedom and movement modes of the arm). To this end, we finetune the backbone model with the given expert videos via Low-Rank Adaptation (LoRA) (Hu et al., 2022), adapting it to the target domain:

$$G' = \text{LoRA}(G; \mathcal{V}, prompt), \qquad (1)$$

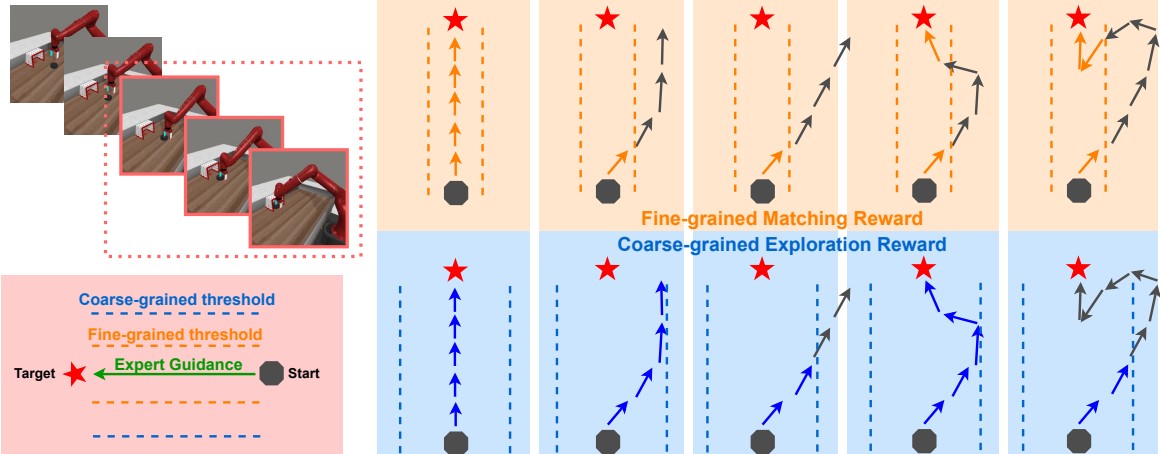

*Figure 2.* The effect diagram of coarse-grained and fine-grained rewards. The top-left panel shows a schematic of the expert trajectory for the *plate slide* task. We take the 2D trajectory arm pushing the plate on a plane in the latter half of the task as an example (bottom-left), with the expert guidance, coarse-grained threshold, and fine-grained threshold represented in different colors. The right panel illustrates how different trajectories trigger the two rewards, respectively. Coarse-grained rewards use a larger threshold to guide sequential target imitation, which encourages the robotic arm to roughly mimic the movement intent in the guidance. However, (i) the larger threshold tolerates operations with deviations, making it hard to learn precise interactions (target reaching here); (ii) its sequential imitation goals mean that trajectories that deviate at first but later correct and even succeed will no longer receive rewards. Fine-grained rewards directly tackle these two problems: they not only prioritize rewarding precise interactions to further refine the policy, but also ensure that trajectories achieving final success without sequential imitation receive positive feedback, thus reducing agent confusion.

where the prompt is corresponding to the specific task. The finetuned video generator $G'$ serves as an expert guide in RL process, generating personalized guidance $V'$ tailored to each RL episode:

$$V' = G'(o_0; prompt), \tag{2}$$

where $o_0 \sim d_0$ denotes the initial observation of the current episode. $V'$ consists of several generated frames $\{v'_i\}_{i=1}^l$, where $l$ denotes the fixed length of generated videos.

### 3.2. Self-supervised Latent Alignment

After obtaining the expert guide, DEG aims to drive the agent to match the generated expert video sequentially to complete the target task, and this can be achieved by rewarding the distance variation between the current observation and the generated video. Considering that the distance calculated directly in the raw pixel space cannot align with the actual semantic distance (Liu & Abbeel, 2021; Yarats et al., 2021), a natural approach is to leverage open-source pre-trained encoders trained on large-scale datasets (Siméoni et al., 2025), computing the distance in the semantic latent space. However, the generated videos inevitably contain noise, and we find that pre-trained models are highly sensitive to such noise, failing to align noisy generated images (e.g., blurry or distorted ones) with their semantically identical real observations. To address this issue, DEG employs noise-alignment-based self-supervised learning to train an encoder $E$ tailored to the target domain, seeking to ignore the non-semantic gaps and map the semantic distance to the computable latent distance.

DEG employs contrastive learning (Stooke et al., 2021; Laskin et al., 2020) to align expert video frames that are manually added with noise yet semantically consistent and push away different frames. Concretely, a batch of images $\{v_i\}_{i=1}^M$ is sampled from the expert video set $\mathcal{V}$. Each $v_i$ is randomly augmented with noise (random shift performs well enough) to obtain two noisy images. They are separately encoded by $E$ and its momentum copy $E'$ (updated by Exponential Moving Average (EMA) (He et al., 2020)) to obtain two latent features, $\hat{z}_i$ and $\check{z}_i$. Note that $E$ consists of both a Convolutional Neural Network (CNN) for image understanding and a Multi-Layer Perceptron (MLP) for self-supervised projection. The formula for contrastive learning is given as follows:

$$\mathcal{L}_{align} = -\log \frac{\exp(u(\hat{z}_i)^\top W \check{z}'_i)}{\sum_{j=1}^M \exp(u(\hat{z}_i)^\top W \check{z}'_j)}, \tag{3}$$

where $u(\cdot)$ is another MLP set to introduce asymmetry for avoiding collapse to trivial solutions and $W$ is the contrastive matrix that is co-trained with $E$ and $u(\cdot)$.

### 3.3. Dual-granularity Dense Reward

With the encoder $E$ trained via self-supervised learning, both generated guidance and real observations are encoded

into the aligned self-supervised latent space where DEG achieves reward calculation. Concretely, DEG takes the cosine similarity between the agent observations and the generated videos $\{v'_i\}_{i=1}^l$ in the latent space as the metric and reward design basis:

$$S(t,i) = \cos\_sim(E(v'_i), E(o_t)) = \frac{E(v'_i)^\top E(o_t)}{\|E(v'_i)\| \cdot \|E(o_t)\|}$$
(4)

where $d_{t,i}$ denotes the cosine similarity between observation $o_t$ at time $t$ and the generated frame $v'_i$ indexed $i$.

The proposed dual-granularity reward comprises two components: a coarse-grained exploration reward and a fine-grained matching reward. They are both based on distance function $S(t,i)$ while achieving distinct effects through different distance thresholds. The coarse-grained exploration reward uses a loose judgment threshold to encourage the agent to roughly and sequentially imitate the robotic motion trajectory in the expert guidance. This not only prevents the agent from being confined to the single trajectory (there are obviously multiple feasible trajectories for the robotic arm to reach the target object, but the expert guidance video only provides one) but also significantly reduces the learning difficulties and thus improves sampling efficiency. However, relying solely on the coarse-grained reward cannot support the agent to perform fine-grained operations in tasks requiring precise interaction with objects, which motivates us to further design the fine-grained matching reward. With a strict judgment threshold, this reward incentivizes agent observations that sufficiently match any guidance frame that is not reached yet, thus fully rewarding the accurate movements or successful interactions with objects. Through the organic integration of the two components, the dual-granularity reward achieves a balance between sampling efficiency and imitation accuracy. See Figure 2 for a concrete example to grasp our motivation on these two rewards.

**Coarse-grained exploration reward** drives the agent to roughly imitate the generated guidance $\{v'_i\}_{i=1}^l$ in sequence, thereby enabling efficient learning of the general motion intent of the robotic arm and achieving task-relevant intrinsic exploration. In each episode, we maintain a variable $I_{\text{target}}$ to represent the index of the current target frame that the agent needs to approach. This variable is initialized as $I_{\text{target}} \leftarrow 1$ and our coarse-grained exploration reward $\mathcal{R}^{coarse}$ is formulated as the similarity gain after reaching the current observation:

$$\mathcal{R}_{coarse} = S(t, I_{\text{target}}) - S(t-1, I_{\text{target}}).$$
(5)

A coarse-grained threshold $\tau_{coarse}$ is set to determine whether the target index $I_{\text{target}}$ can be sequentially advanced

to the next target by fixed step number $s$:

$$I_{\text{target}} \leftarrow \begin{cases} \min(I_{\text{target}} + s, l), & \text{if } S(t, I_{\text{target}}) > \tau_{\text{coarse}} \\ I_{\text{target}}, & \text{otherwise,} \end{cases}$$
(6)

**Fine-grained matching reward** encourages observations that sufficiently match the expert video, thus rewarding the robotic arm for accurate movements or successful interactions with objects. Considering that (i) the trajectories for reaching the target object can be highly diverse and different with guidance and (ii) the poses required for object interaction are rather consistent and semantically aligned with the corresponding frames in the generated guidance, we don't adopt a strictly sequential target update mechanism, which is employed in coarse-grained reward. Instead, we calculate the cosine similarity between the current observation $o_t$ and the whole guidance $\{v'_i\}_{i=1}^l$, identifying the generated frame that is most similar to the current observation:

$$i^* = \arg\max_{i \in \{1,2,\dots,l\}} S(t,i).$$
(7)

Then, a strict fine-grained threshold $\tau_{\text{fine}}$ is employed to determine whether $o_t$ sufficiently matches $v'_{i^*}$, and then grants a reward accordingly:

$$R_{fine} = \begin{cases} i^*, & \text{if } S(t, i^*) > \tau_{\text{fine}} \quad \& \quad i^* > I_{\text{reached}} \\ 0, & \text{otherwise,} \end{cases}$$
(8)

where $I_{\text{reached}}$ is set to record the rearmost (i.e., largest index) frame that has been successfully matched so far, which prevents the agent from "slacking off" and repeatedly obtaining rewards by matching the same or previous frames.

**Dual-granularity reward** employed in DEG consists of the above two sub incentives with coefficients $\alpha$ and $\beta$:

$$\mathcal{R}_{DEG} = \alpha \mathcal{R}_{coarse} + \beta \mathcal{R}_{fine}.$$
(9)

It can guide effective RL independently or achieve more efficient learning with the help of a success sparse reward $\mathcal{R}_s$ that is easy to obtain and scaled by $\theta$:

$$\mathcal{R}_{DEG}^+ = \alpha \mathcal{R}_{coarse} + \beta \mathcal{R}_{fine} + \theta \mathcal{R}_s.$$
(10)

## 4. Experiment

We conduct extensive experiments to answer the following questions: (Section 4.1) Can DEG, relying solely on

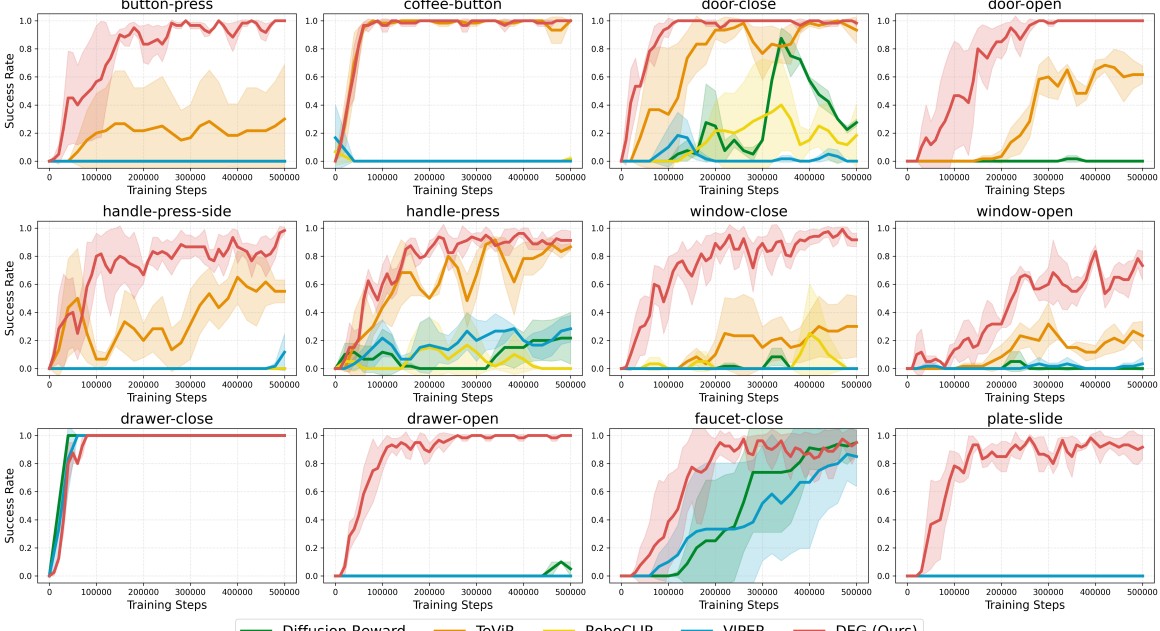

*Figure 3.* Comparison with state-of-the-art reward engineering methods on 12 task-free tasks. DEG achieves better performance on both the final policy level and RL sample efficiency. Owing to the non-open-sourcing and constraints on computational resources, we employ the results of TeViR and RoboCLIP provided by TeViR only for the first eight tasks.

a small amount of expert videos, independently guide effective RL and outperform SOTA baselines (equipped with sufficient multi-view expert data) in terms of sampling efficiency? (Section 4.2) With the aid of successful sparse rewards, can DEG achieve performance comparable to that of handcrafted rewards designed by human experts? (Section 4.3) Can DEG also work in the real-world manipulation domains? (Section 4.4) Are both components of the dual-grained reward effective? (Section 4.5) Compared with large-scale pre-trained encoders, does the self-supervised encoder possess superior latent space alignment capabilities? In addition, we further provide more results and analysis in Appendix B.

### 4.1. Comparison on Reward-free Tasks

In this section, we compare the proposed DEG on reward-free manipulation tasks against several popular and state-of-the-art baselines: TeViR (Chen et al., 2025a), Diffusion Reward (Huang et al., 2024), Viper (Escontrela et al., 2024), and RoboCLIP (Sontakke et al., 2023). Following these works, we conduct experiments on 12 challenging and diverse Metaworld manipulation tasks (Yu et al., 2020; McLean et al., 2025). These tasks cover a diverse and challenging set of robotic arm operations, essentially spanning the entire feasible state space of the Sawyer robotic arm. DrQv2 (Yarats et al., 2022) is employed as the backbone for all methods, with only 500k steps allowed to fully evaluate the RL sample efficiency. See Appendix A for detailed

description of baseline methods and manipulation tasks.

For each task, DEG is only provided with a small number of expert videos (3-5 clips), and no other expert supervision or human annotations are used. Considering the different data requirements of the baselines, we conduct experiments according to their own settings or directly use the results from previous papers. For example, the TeViR (Chen et al., 2025a) relies on 60 videos and requires three camera viewpoints (the left side of the arm, the top of the arm, and a close-up view of the gripper). We follow its requirements for TeViR, and the same applies to other methods. Note that although we present all results together for comparison, our DEG is still only trained with a few videos with a single camera.

The results are shown in Figure 3. Except for *door-close* where DEG and TeViR achieve similar final performance, and *drawer close* where all baselines converge well, DEG consistently and significantly outperforms all baselines on the remaining 10 tasks. This demonstrates DEG's ability to independently guide effective reinforcement learning and ultimately drive stable policy convergence. Note that TeViR uses sufficient expert videos from three viewpoints, while DEG only uses a small number of single-view videos. In terms of sample efficiency, DEG also shows a huge advantage across all tasks, which proves the high precision of the proposed dual-grained self-supervised dense reward in the semantic guidance of robotic arm movements.

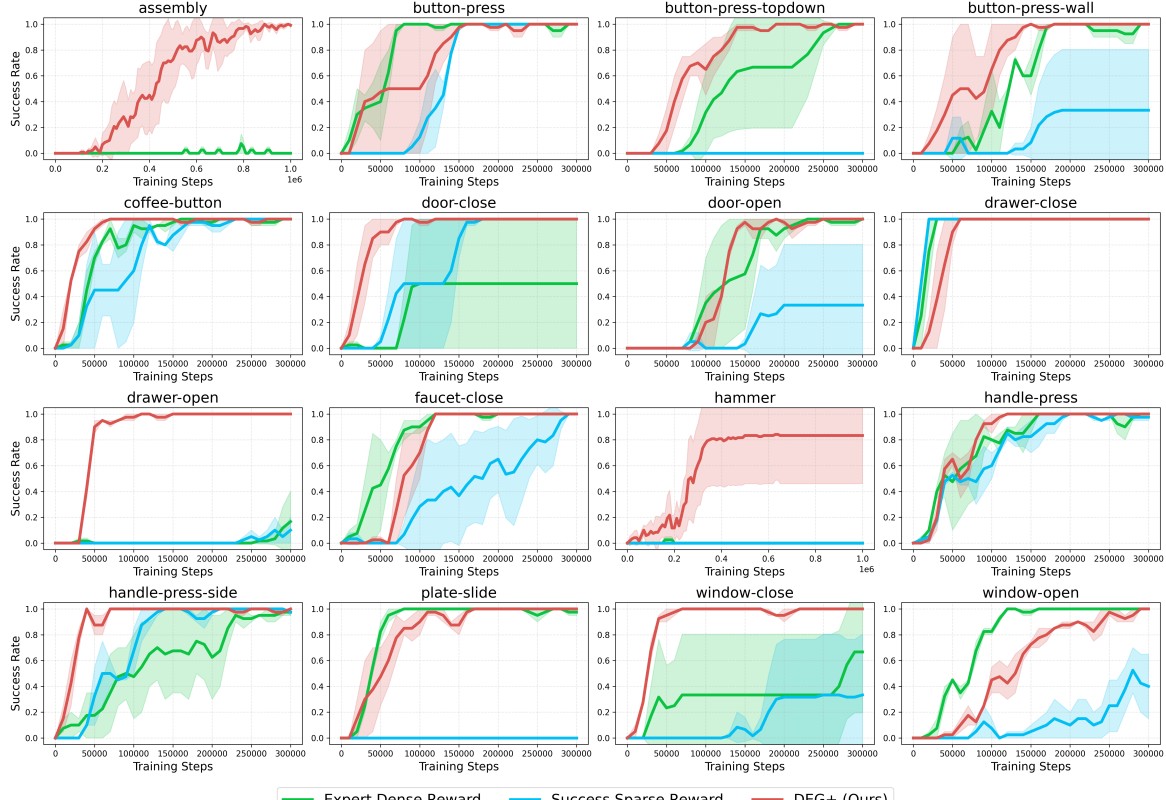

*Figure 4.* With success sparse reward, DEG+ can effectively improve RL efficiency and match human expert-annotated dense reward. It even outperforms across several tasks, such as *assembly*, *hammer*, and *drawer-open*.

### 4.2. DEG with Sparse Success Feedback

In this section, we demonstrate the performance of DEG with the help of an easy-to-obtain success sparse reward, named DEG+. To demonstrate the effectiveness of DEG, we directly compare it with the latest human expert-annotated dense reward (McLean et al., 2025) on the MetaWorld benchmark, and refer to this method as Expert Dense Reward. In addition, the RL results of Success Sparse Reward (Yu et al., 2020) are also used for comparison.

The results across all 16 tasks are shown in Figure 4. Without dense incentives, Success Sparse Rewards struggle to guide stable RL. This is because the exploration space in long-horizon tasks is vast, and it is hard for the agent to complete the task and obtain signals through the exploration provided by the randomness of the stochastic policy. In contrast, DEG provides an exclusive successful trajectory for each RL episode and delivers efficient guiding rewards accordingly, which greatly improves the efficiency of exploring sparse rewards. In addition, even when compared with the dense rewards handcrafted by human experts, DEG can still match their performance on most tasks and outperform them on several tasks, mainly due to the complete information provided by expert guidance videos. Take *drawer*

*open* as an example: the expert dense reward incentivizes the distance between the gripper and the handle, which can cause the gripper to push against the handle from the outside, making it hard to open the drawer, falling into a local optimum. DEG, however, can accurately learn from the generated expert guidance to lift the gripper, aim at the inner side of the handle and then lower it, achieving extremely stable policy convergence.

### 4.3. Real-world Experiments

To further explore the capabilities of DEG, we conduct experiments on physical robotic arm manipulation tasks. We design two tasks on the Franka robotic arm: stack cube and pick banana. Both are long-horizon tasks that require precise interaction between the robotic arm and objects. You can refer to Appendix A for the detailed description of the real-world tasks. We adopt the state-of-the-art human-in-the-loop algorithm, HIL-SERL (Luo et al., 2025a), as the RL backbone of the physical RL algorithm and allow a maximum of 25k steps for each task.

The results are shown in Figure 5. DEG can also provide real-world manipulators with effective task-level incentives, significantly improving the sampling efficiency

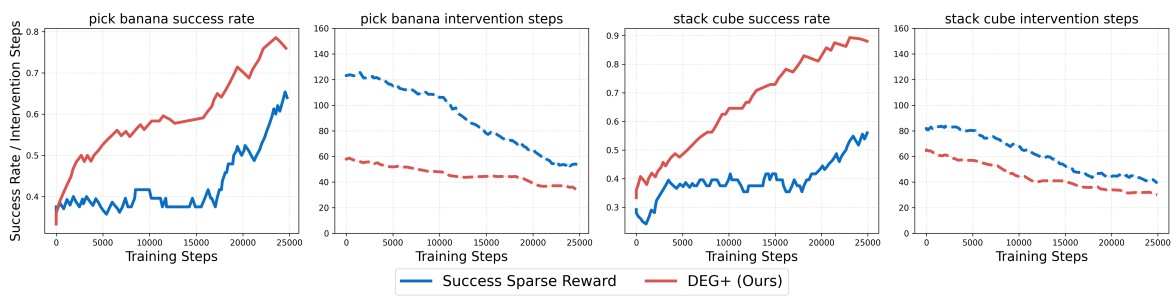

*Figure 5.* Results on real-world franka manipulation tasks. DEG enables better RL efficiency and lower intervention rates than employing only success sparse reward.

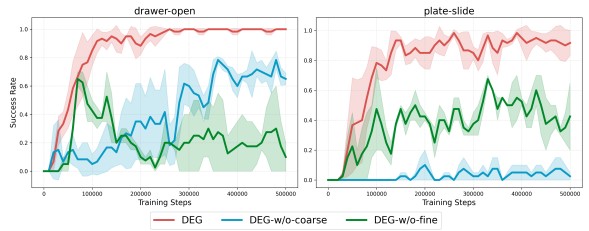

*Figure 6.* Numerical ablation experiments. Both coarse-grained reward and fine-grained reward are useful in DEG.

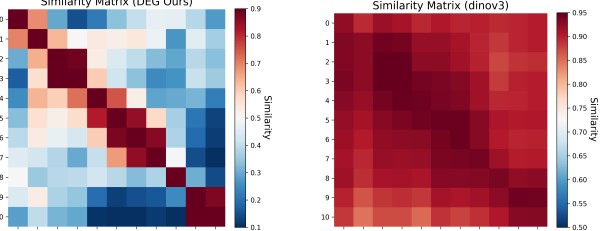

*Figure 7.* Frame similarity heatmap between generated videos and real videos with identical trajectories (*drawer-open*). The horizontal axis represents the generated frame sequence (interval 5), and the vertical axis represents the real observation sequence. Compared with DINO-v3 (right), the self-supervised encoder (left) can better align frames with the same semantics and distinguish those with different semantics.

of real-world RL. In addition, we observe that DEG can generate abundant dense incentives when humans provide help, thereby rapidly reducing the intervention rate. Since object states are not directly accessible in the real world as they are in simulators, handcrafting dense rewards becomes extremely difficult, while DEG can realize effective dense reward design for physical tasks with very easy data collection.

### 4.4. Ablation Study

In this section, we provide ablation experiments on the proposed dual-granularity dense reward, verifying whether both sub-rewards are useful. Experimental results (Figure 6) confirm that both contribute to DEG's achievement of sample-efficient reinforcement learning. In addition, we further provide the visualization of the policy intentions learned by DEG and its two ablation variants on the drawer open task, shown in Figure 9 in Appendix B. With only the coarse-grained reward, DEG can quickly learn the overall motion trajectory of the robotic arm in the expert videos but fails to achieve precise interaction with the drawer. With only the fine-grained reward, the robotic arm can grasp the handle yet lacks the tendency to imitate the motion trajectory.

### 4.5. Encoder Analysis

In DEG, we train the reward calculation encoder via self-supervised learning, instead of adopting state-of-the-art pre-trained models such as DINOv3 (Siméoni et al., 2025). In this section, we visualize the matching performance of different encoders when facing real and generated videos, as

shown in the Figure 7. Although pre-trained encoders can adapt to different tasks via downstream predictors, their extensive prior knowledge makes minor visual variations in a single target domain less distinguishable in the latent space. The self-supervised encoder can align semantically identical frames and distinguish semantically different ones, and map the semantic distance to the similarity in the latent space, thus enabling DEG to design dual-grained rewards based on this similarity.

## 5. Conclusion & Limitation

In this paper, we propose DEG, a sample-efficient reward design framework for embodied manipulation tasks without requirements on human annotations or extensive supervision. Via video generation models as a RL guide, the proposed dual-granularity reward effectively encourages the agent to imitate the generated guidance in the self-supervised space, achieving efficient RL and surpassing state-of-the-art baselines. The main bottleneck of DEG lies in the slow video generation speed. Since pre-training data for video generation models is relatively clear, reducing the resolution to speed up generation may cause the collapse. We believe this issue will likely be resolved with the development of foundation models, and DEG can inspire more researchers to focus on data-efficient dense reward design.

## Impact Statement

This paper presents work whose goal is to advance the field of Machine Learning. There are many potential societal consequences of our work, none which we feel must be specifically highlighted here.

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

# A. Experimental Details

## A.1. Task Description

Following previous works (Huang et al., 2024; Chen et al., 2025a; Liu et al., 2025c), we employ sixteen simulation tasks from Metaworld (Yu et al., 2020; McLean et al., 2025), they are: *door-close*, *door-open* , *window-close* , *window-open* , *handle-press* , *handle-press side* , *button-press* , *coffee-button* , *drawer-open* , *faucet-close* , *plate-slide* , *drawer-close* , *button-pres-topdown* , *button-press-wall* , *hammer* , *assembly*. These tasks cover a diverse and challenging set of robotic arm operations, essentially spanning the entire feasible state space of the Sawyer robotic arm. For real-world tasks, we employ franka arm as the manipulator, designing two tasks: *pick-banana* and *stack-cube*. We provide the description below:

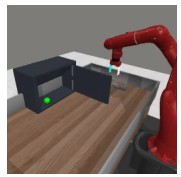

*door-close.*
Closing a door mounted on a revolving hinge joint. The task requires adapting the pushing trajectory to account for the door's randomized initial angles.

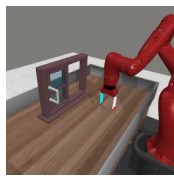

*window-close.*
Executing a push motion to close a sliding window, starting from a randomized open position along its rail.

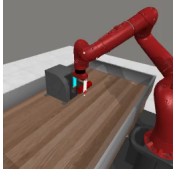

*handle-press.*
Depressing a lever handle to its endpoint. The handle's initial orientation and height are varied.

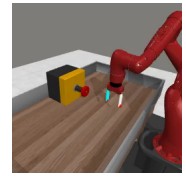

*button-press.*
Actuating a standard push-button from a randomized location in the workspace.

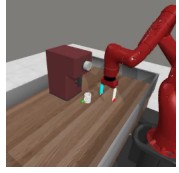

*coffee-button.*
Pushing a button on a coffee machine to trigger dispensing. The position of the coffee machine is randomized.

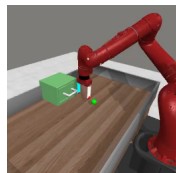

*drawer-open.*
Grasping and pulling open a drawer from a randomized closed position.

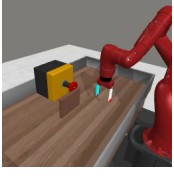

*button-press-wall.*
A navigation and manipulation task that involves bypassing a physical barrier to reach and press a button located behind it, with randomized relative positioning.

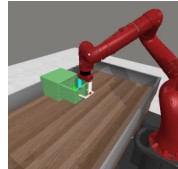

*drawer-close.*
Pushing a drawer shut from a randomized open displacement.

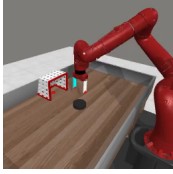

*plate-slide.*
Sliding a plate laterally into a constrained cabinet opening. The initial placements of the plate and the target cabinet are varied.

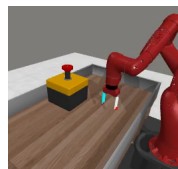

*button-press-topdown.*
Pressing a button via a precise top-down approach, with randomized button placement.

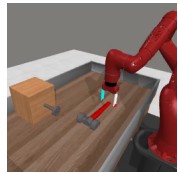

*hammer.*
A dynamic task requiring the accurate swinging of a hammer to strike the head of a wall-mounted screw. Randomization of both tool and target positions assesses swing trajectory planning and impact control.

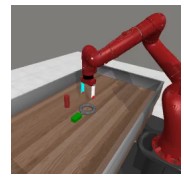

*assembly.*
A bi-manipulation task involving the sequential picking of a free nut and its precise placement onto a fixed peg. Randomization of both object positions tests fine grasping, alignment, and insertion skills.

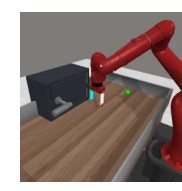

*door-open.*
Opening a hinged door from a randomized closed position, involving grasping and pulling a handle from variable configurations.

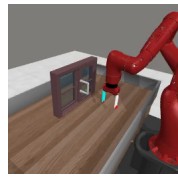

*window-open.*
Opening a sliding window by pushing it from a randomized initial closed or ajar state.

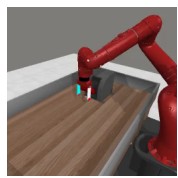

*handle-press-side.*
Applying downward force on a handle along a sideways axis, with randomized handle poses.

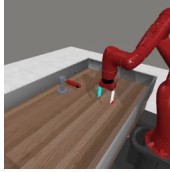

*faucet-close.*
Rotating a faucet knob clockwise to its closed position, starting from a randomized angular position.

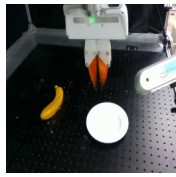

*Real-world: pick-banana.*
Grasping a banana from a randomized initial position and placing it onto a fixed-location white plate. The task evaluates precision grasping of deformable objects and targeted placement.

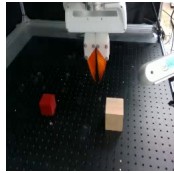

*Real-world: stack-cube.*
Stacking a small red cube from a randomized start position onto a larger, fixed-position natural wood-colored cube. The task assesses spatial alignment and stable placement for basic assembly.

### A.2. Baseline Introduction

In Section 4.1, we employ four popular and advanced baselines: TeViR (Chen et al., 2025a), Diffusion Reward (Huang et al., 2024), VIPER (Escontrela et al., 2024), and RoboCLIP (Sontakke et al., 2023).

**TeViR** (Chen et al., 2025a) utilizes a conditional video diffusion model to generate the multi-view trajectory of distance-based reward calculation. It is trained under three different cameras.

**Diffusion Reward** (Huang et al., 2024) leverages a conditional video diffusion model to capture the expert demonstration distribution and uses the conditional entropy of the video prediction with the sparse reward to formulate dense rewards.

**VIPER** (Escontrela et al., 2024) utilizes a video prediction transformer and directly leverages the model's prediction likelihood as a reward signal to encourage the agent to learn the expected behavior.

**RoboCLIP** (Sontakke et al., 2023) utilizes pre-trained Vision-Language Models (VLM) to generate rewards by calculating the similarity of the agent's observation to the language task description for the agent.

In Section 4.2, we use the Success Sparse Reward and the Expert Dense Reward provide by the Metaworld environment (Yu et al., 2020; McLean et al., 2025).

**Success Sparse Reward** (Yu et al., 2020) is corresponding to the success signals provided by the environments. It is is proportional to info['success'] in standard Metaworld environment.

**Expert Dense Reward** (McLean et al., 2025) is the latest dense reward designed by experts on the MetaWorld benchmark, which can guide RL (Schulman et al., 2017) to achieve effective policy learning within 20M steps. In contrast, DEG can complete most tasks within 300k steps with the aid of sparse success rewards.

### A.3. Hyper-parameters

We provide the hyper-parameters of DEG, as shown in 1. The information (e.g., prompt) in large model training is provided in the next section.

*Table 1.* The hyper-parameters of DEG. For domain *door*, *coffee*, and task *drawer-close*, the ($I_{coarse}$, $I_{fine}$) is (0.8, 0.95); For domain *window*, the ($I_{coarse}$, $I_{fine}$) is (0.8, 0.96); for *faucet*, $I_{coarse}$ is 0.9. For detailed number of used videos, please refer to the next section.

| Hyper-parameter | Setting |
| --- | --- |
| Video fondation Model | Wan2.1-I2V-14B |
| Number of finetuning videos | 3-5 (details in next section) |
| Video frames | 250 |
| Frame shape | $256 \times 256 \times 3$ |
| Frame stack | 1 |
| Action repeat | 2 |
| Action type | Continuous |
| Action dimension | See (McLean et al., 2025) |
| Large model finetuning | LoRA |
| self-supervised pre-training times | 5000 |
| Feature encoder lr | $1e-4$ |
| Random shift (noise) upper bound | 4 |
| EMA update frequency | 2 |
| EMA Momentum | 0.05 |
| Optimizer | Adam (Kingma & Ba, 2014) |
| Coarse-grained coefficient $\alpha$ | 100 |
| Coarse-grained threshold $I_{coarse}$ | 0.85 |
| Coarse-grained frequency $s$ | 4 |
| Fine-grained coefficient $\beta$ | 1 |
| Fine-grained threshold $I_{fine}$ | 0.98 |
| Sparse reward weight | 10 (if used) |
| Metaworld RL backbone | DrQv2 |
| Discount factor | 0.98 |
| Batch size | 256 |
| n-step returns | 3 |
| Random frames in RL | 4000 |
| Feature dimension | 50 |
| Task difficulty for exploration | Hard |
| Other DrQv2 hyper-parameters | See (Yarats et al., 2022) |
| Real-world RL backbone | HIL-SERL |
| HIL-SERL hyper-parameters | See (Luo et al., 2025a) |

### A.4. Finetuning Details: Framework, Data, and Prompt

In DEG, we employ LoRA (Hu et al., 2022) to finetune Wan2.1-I2V-14B (Wan et al., 2025) as our RL guide. We use DiffSynth-Studio `https://github.com/modelscope/DiffSynth-Studio` as our finetuning framework, following its default settings on Wan2.1-I2V-14B. The number of videos and task prompt employed in each task is provided here.

---

*door-close (3 videos)*:

A video clip of the door-close task: A red robotic arm is positioned on a table, along with a dark cabinet whose door is initially open (with its position randomly initialized and then fixed), and a light green target marker. The robotic arm moves toward the cabinet door, presses against it from the outside, and closes it. Keep the table static and the cabinet position unchanged throughout. Preserve detailed shadows on the ground. Ensure that the motion of the robotic arm and the cabinet door follows logical physics. Maintain consistent colors and appearance of objects and the scene from start to finish. After initialization, the cabinet door exhibits a slight automatic rebound, then remains stationary until contacted by the robotic arm.

---

*window-close (3 videos)*:

A video clip of the window-close task: A red robotic arm is positioned on a table alongside an open window. The robotic arm moves toward the window, presses against its white handle from the left side, and slides it to the right to close the window. Keep the table static and the window position unchanged throughout. Preserve detailed ground shadows. Ensure that the motion of the robotic arm and the window follows logical physics. Maintain consistent colors and appearance of the scene and objects from start to finish.

---

*handle-press (5 videos)*:

A video clip of the handle-press task: A red robotic arm is positioned on a table, along with a device featuring a red handle (the device position is randomly initialized and then fixed). The robotic arm lifts up, moves above the handle, then descends and uses its gripper to press the handle downward onto the table. Keep the table static and the device position unchanged throughout. Preserve detailed ground shadows. Ensure that the motion of the robotic arm follows logical physics, and maintain consistent colors and shapes of the objects and scene from start to finish. Note that the handle does not move until the robotic arm makes contact with it.

---

*button-press (3 videos)*:

A video clip of the button-press task: A red robotic arm is positioned on a table, along with a box featuring a red button. The robotic arm moves toward the box, accurately aligns with the red button, and presses it down. Keep the table static and the box position unchanged throughout. Preserve detailed ground shadows. Ensure that the motion of the robotic arm follows logical physics, and maintain consistent colors and shapes of the objects and scene from start to finish.

---

*coffee-button (5 videos)*:

A video clip of the coffee-button task: A red robotic arm is positioned on a table, along with a coffee machine with a button (machine position randomly initialized and then fixed), and a coffee mug. The robotic arm reasonably adjusts its orientation until aligned with the coffee machine, then moves toward the button and presses it. Keep the table static and the coffee machine position unchanged throughout. Preserve detailed ground shadows. Ensure that the motion of the robotic arm follows logical physics, and maintain consistent colors and shapes of the objects and scene from start to finish. Note that the button does not move until the robotic arm makes contact with it.

---

*drawer-open (3 videos)*:

A video clip of the drawer-open task: A red robotic arm is positioned on a table, along with a closed green drawer featuring a white handle. The robotic arm lifts up and moves above the white handle, then descends and uses its gripper to precisely hook the handle (with the blue inner jaw positioned inside and the white outer jaw positioned outside), and pulls the drawer open. Keep the table static. Preserve detailed ground shadows. Ensure that the motion of the robotic arm and the target object follows logical physics, and maintain consistent colors and shapes of the objects and scene from start to finish. Note that the drawer does not move until the robotic arm makes contact with it or its handle.

*button-press-wall (3 videos)*:

A video clip of the button-press-wall task: A red robotic arm is positioned on a table, along with a box featuring a red button and a fixed wall. The robotic arm lifts up and closes its gripper, moves toward the box and passes over the wall, then descends to align the outer side of its gripper precisely with the red button, opens the gripper, and presses the button down. Keep the table static and the box position unchanged throughout. Preserve detailed ground shadows. Ensure that the motion of the robotic arm follows logical physics, and maintain consistent colors and shapes of the objects and scene from start to finish. Note that the red button does not move until the robotic arm makes contact with it, and the robotic arm cannot pass directly through the wall.

*drawer-close (3 videos)*:

A video clip of the drawer-close task: A red robotic arm is positioned on a table, along with an open green drawer featuring a white handle. The robotic arm first lifts upward, then descends and uses its gripper to precisely press against the white handle from the outside, pushing the green drawer closed. Keep the table static. Preserve detailed ground shadows. Ensure that the motion of the robotic arm and the target object follows logical physics, and maintain consistent colors and shapes of the objects and scene from start to finish. Note that the drawer does not move until the robotic arm makes contact with it or its handle.

*plate-slide (3 videos)*:

A video clip of the plate-slide task: A red robotic arm is positioned on a table, alongside a black plate initialized at a random position, and a soccer goal. The robotic arm moves downward, accurately presses on the plate, and slides it into the goal toward a red target point. Keep the table and goal static with unchanged positions. Preserve detailed shadows on the ground. Ensure the motion of the robotic arm is logical and that the colors and shapes of the objects and scene remain consistent. Note that the plate does not move until the robotic arm makes contact with it.

*button-press-topdown (3 videos)*:

A video clip of the button-press-topdown task: A red robotic arm is positioned on a table, along with a box featuring a red button on its top surface. The robotic arm lifts upward, then moves over the box, aligns with the red button, and descends to press it down precisely. Keep the table static and the box position unchanged throughout. Preserve detailed ground shadows. Ensure that the motion of the robotic arm follows logical physics, and maintain consistent colors and shapes of the objects and scene from start to finish. Note that the red button does not move until the robotic arm makes contact with it.

*hammer (3 videos)*:

A video clip of the hammer task: A red robotic arm is positioned on a table, along with a hammer featuring a red handle and a wooden block with a nail. The robotic arm moves toward the hammer, uses its gripper to precisely grasp the hammer's red handle, lifts the hammer, moves it to align the hammerhead with the nail, and presses the nail into the wooden block. Keep the table static and the wooden block position unchanged throughout. Preserve detailed ground shadows. Ensure that the motion of the robotic arm and the hammer follows logical physics, and maintain consistent colors and shapes of the objects and scene from start to finish.

*assembly (3 videos)*:

A video clip of the assembly task: A red robotic arm is positioned on a table, along with a circular part featuring a green handle and a fixed red cylinder. The robotic arm moves toward the part, uses its gripper to precisely grasp its green handle, lifts the part, moves it toward the red cylinder, and places the circular part onto the cylinder. Keep the table static and the red cylinder position unchanged throughout. Preserve detailed ground shadows. Ensure that the motion of the robotic arm and the part follows logical physics, and maintain consistent colors and shapes of the objects and scene from start to finish.

*door-open (3 videos)*:

A video clip of the door-open task: A red robotic arm is positioned on a table, along with a black cabinet with a gray handle (cabinet position randomly initialized and then fixed), and a light green target marker. The robotic arm lifts up, moves toward the cabinet handle while gradually closing its gripper, uses the gripper to precisely press against the gray handle of the black cabinet from the inside, then pulls the cabinet door open until it reaches the position of the green target marker. Keep the table static and the cabinet position unchanged throughout. Preserve detailed ground shadows. Ensure that the motion of the robotic arm and the cabinet door follows logical physics, and maintain consistent colors and appearance of the objects and the scene from start to finish.

*window-open (5 videos)*:

A video clip of the window-open task: A red robotic arm is positioned on a table, along with a closed window whose position is randomly initialized and then fixed. The robotic arm lifts up, moves toward the window, uses its gripper to press against the window's white handle from the right side, and then moves leftward to push the window open. Keep the table static and the window position unchanged throughout. Preserve detailed ground shadows. Ensure that the motion of the robotic arm and the window follows logical physics, and maintain consistent colors and appearance of the scene and objects from start to finish.

*handle-press-side (3 videos)*:

A video clip of the handle-press-side task: A red robotic arm is positioned on a table, along with a dark gray device featuring a red handle (device position randomly initialized and then fixed). The robotic arm lifts up, moves above the handle, then descends and presses the handle down. Keep the table static and the device position unchanged throughout. Preserve detailed ground shadows. Ensure that the motion of the robotic arm follows logical physics, and maintain consistent colors and shapes of the objects and scene from start to finish with no pixelation or artifacts. Note that the handle does not move until the robotic arm makes contact with it.

*faucet-close (3 videos)*:

A video clip of the faucet-close task: A red robotic arm is positioned on a table, along with a faucet featuring a red knob. The robotic arm moves toward the faucet, uses its gripper to precisely press against the red knob from the left side, and rotates it clockwise to close the faucet. Keep the table static and the faucet position unchanged throughout. Preserve detailed ground shadows. Ensure that the motion of the robotic arm and the target object follows logical physics, and maintain consistent colors and shapes of the objects and scene from start to finish. Note that the knob does not move until the robotic arm makes contact with it.

*real-world: pick banana (3 videos)*:

A video clip of the robotic arm successfully grasping and placing a banana into a plate. On a table, there are a plate, a banana initialized at a random position, and a robotic arm equipped with a gripper. The robotic arm moves horizontally above the banana, then opens its gripper, descends to precisely grasp the banana with the gripper, lifts it, moves horizontally above the plate, opens the gripper, and places the banana into the plate.

> *real-world: stack cube (3 videos)*:
>
> A video clip of the robotic arm stacking a cube. On a table, there are a plate, a red cube initialized at a random position, and a wooden block. The robotic arm moves horizontally above the red cube, then opens its gripper, descends to precisely grasp the red cube with the gripper, lifts it, moves horizontally above the wooden block, opens the gripper, and places the red cube onto the wooden block.

## B. Additional Results & Analysis

### B.1. More Ablation Results

Ablation experiments on three reward-free tasks and one sparse-reward task (marked with DEG+). The results in Figure 8 further demonstrate that both sub rewards are useful in the proposed dual-granularity reward.

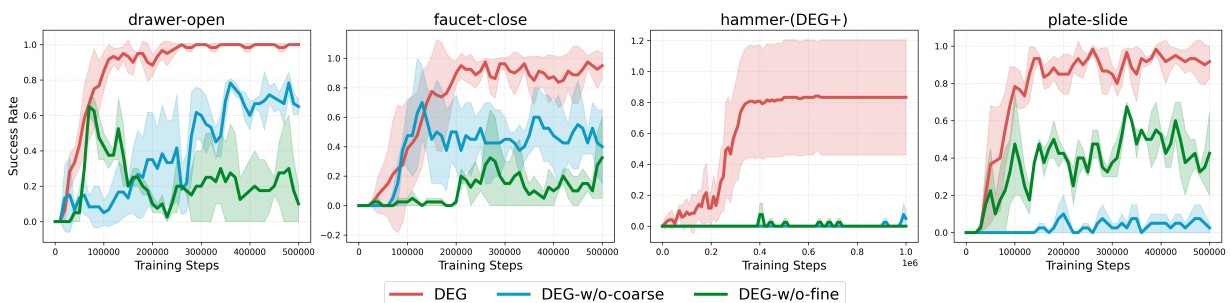

*Figure 8.* Numerical ablation study. Both coarse-grained and fine-grained rewards are necessary.

### B.2. Ablation: Policy Intention Visualization

In addition to the numerical comparison, we further provide the visualization of the policy intentions learned by DEG and its two ablation variants. We employ the learned policy in *drawer-open* without success sparse reward at 250k steps and show the results in the Figure 9. With only the coarse-grained reward, DEG can quickly learn the overall motion trajectory of the robotic arm in the expert videos but fails to achieve precise interaction with the drawer. With only the fine-grained reward, the robotic arm can grasp the handle yet lacks the tendency to imitate the motion trajectory.

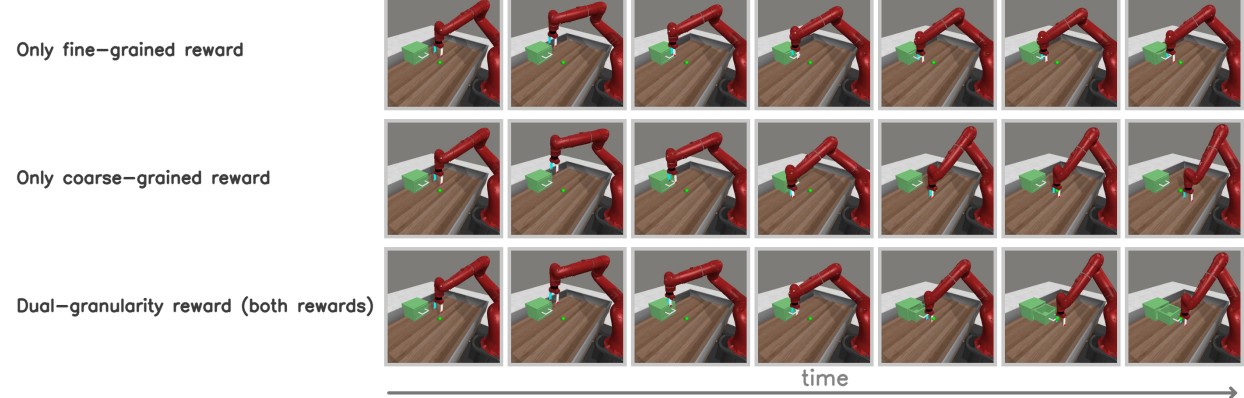

*Figure 9.* Policy Intention Visualization of DEG and its two ablation variants. Each of the two sub-rewards has its own strengths and weaknesses, while the dual-granularity reward can combine the advantages of both.

### B.3. Generated Episodic Guidance in DEG

In this chapter, we present a direct comparison between the generated videos and real videos used in DEG, shown in Figure 10 and Figure 11. We find that the fine-tuned video generation model can achieve effective domain adaptation. Despite remaining noise in terms of clarity and certain details, it basically recovers the semantic manner of real task execution. Note that real videos (3-5 clips) are only used to fine-tune the video generation model, and DEG uses generated videos to calculate rewards throughout the entire RL process.

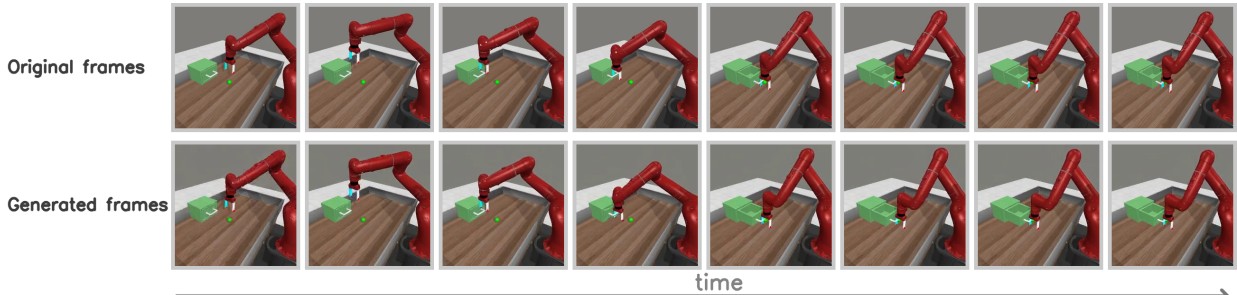

*Figure 10.* DEG generated videos versus real videos on Metaworld *Drawer-open* task.

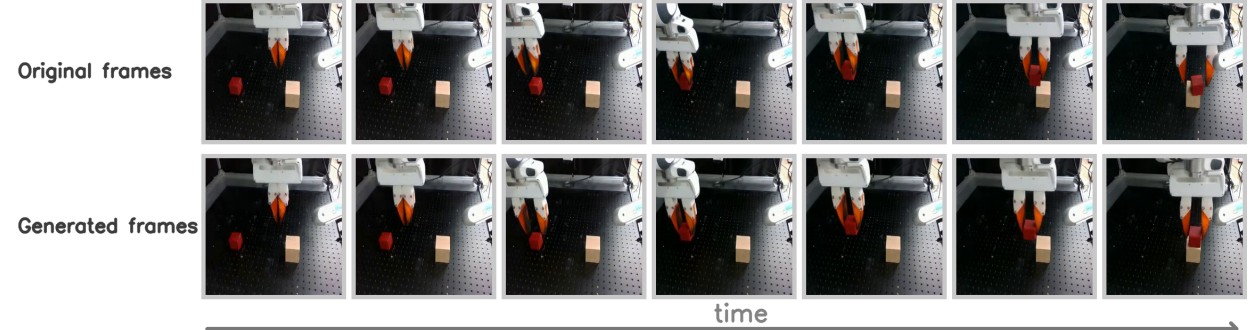

*Figure 11.* DEG generated videos versus real videos on real-world *stack-cube* task.

