# OpenReview forum: "Dual-granularity Dense Reward over Generated Episodic Guidance for Embodied Reinforcement Learning"
_ICML.cc/2026/Conference — Submitted to ICML 2026_

### Official Review · Reviewer_5RPe · 2026-03-07

**Soundness:** 3
**Presentation:** 2
**Significance:** 2
**Originality:** 3
**Overall Recommendation:** 4
**Confidence:** 4

**Summary:**

Reward design remains a long-standing challenge in embodied robotic manipulation. This paper tries to address this problem by proposing DEG, a reward design framework that leverages video generation models to produce reward signals from a few expert demonstrations. The core components of DEG include a self-supervised latent alignment scheme to mitigate the adverse effects of inevitable noise in generated frames, as well as a dual-granularity reward mechanism that jointly encourages exploration and imitation. Experimental results in both simulation and real-world settings demonstrate that DEG effectively improves robotic policy training in terms of  success rates and sample efficiency.

**Compliance With Llm Reviewing Policy:**

Affirmed.

**Final Justification:**

During the rebuttal phase, authors have done substantial work to address the other reviewers' and my concerns. Now all my concerns and follow-up questions have been solved. I believe this paper deserves a positive score. Thus, I would like to raise my final recommendation score to 4.

**Key Questions For Authors:**

1. What is the RL algorithm used in the simulation experiments?
2. How many trials are used to compute the success rates?
3. How are the expert videos acquired?
4. How to draw the training curves in Figure 3-5? Is it done by training K steps and testing a epoch? If so, how large is K?
5. Although DEGs show significant improvement over baselines in terms of training steps, I wonder whether its advantages still persist when reporting the actual training time.

**Limitations:**

Please refer to the major concerns in "Strengths and Weaknesses" for details.

**Strengths And Weaknesses:**

**Strengths:**

1. The design of DEG is comprehensive and addresses several core challenges in using video generation models for robotic policy training. For instance, it proposes a self-supervised latent alignment approach to mitigate negative effects caused by the inevitable noise in the generated frames, and proposes two complementary reward components to balance the exploration and exploitation.
2. Experimental results demonstrate that DEG significantly improves sample efficiency, and the real-world experiments confirm its ability to generalize effectively to physical robotic settings.

**Weaknesses:**

***Major concerns:***

1. The presentation of this paper can be improved in the following aspects:
   - Figure 2 requires revision. It is difficult to extract useful information from the current version even with the lengthy caption. Specifically, it is hard to capture the design rationale behind the two proposed rewards or understand how they function. Although the right panel depicts both rewards, their visual representations are almost the same and only differ in colors (e.g., arrow colors). Yet, the current figure lacks annotations to explain what these colors means, which makes it difficult for readers to understand the core design of DEG.
   - The presentation of experiment results should be improved. At present, all experimental illustrations can be regarded as qualitative analysis. The current manuscript does not provide any quantitative analysis, especially  the final success rates of each method across the benchmarked tasks. One straightforward improvement would be to include a comprehensive summary table that reports the performance of all methods on each task. This will provide readers with a clear and comparable overview of the effectiveness of each method.
2. The scalability of DEG is concerning.
   - First, DEG requires task-specific fine-tuning of the pre-trained video generation model using downstream expert videos to provide effective guidance for policy training. This necessitates collecting different expert videos and training separate LoRA weights for each task. In consequence, both the number of expert videos used for fine-tuning and the training overhead grow linearly with the number of downstream tasks, raising doubts on the scalability of DEG.
   - Second, the reward design in DEG introduces many hyperparameters (five in total, i.e.,   $\tau_{coarse}, \tau_{fine},\alpha, \beta, \theta$). As a result, it may still require extensive hyperparameter tuning to achieve strong performance in policy learning. This somewhat diminishes the advantages of DEG over traditional reward engineering approaches, especially with respect to reducing manual reward design efforts and associated costs.
3. The current manuscript overlooks a number of relevant prior works. Reward densification for robotic manipulation is an active area with many research works (including but not limited to [a, b, c, d]). Authors should at least discuss how DEG differs from existing approaches in the Related Work section to better position DEG within the current literature.
4. More experiments can be conducted to better understand the effectiveness and limitations of DEG.
   - Regarding reward design for robotic manipulation, there are many important works that can be included as baselines in Section 4.1 or 4.2, such as VLAC [a], Robo-dopamine, VLA-RFT [c], and NORA 1.5 [d]. In particular, like DEG, VLA-RFT and NORA 1.5 also uses video generation models (or world models) to provide supervised signals for policy training, making them particularly suitable for comparison.
   - Since DEG is highly dependent on the quality of expert videos, it would be helpful to understand its capability boundary by investigating how the quantity and quality of expert videos affect the performance of DEG. Unfortunately, the current manuscript lacks such in-depth analysis. Additionally, conducting experiments to examine the impact of the hyperparameters in the reward functions in Eq. (9) and Eq (10) would further help readers understand the robustness of DEG.
   - All simulation experiments appear to be conducted under a single-task training paradigm. To better evaluate the generalization and practical applicability of each method, it is suggested to explore multi-task co-training. This can be done by grouping related tasks into task suites just like Libero benchmark.
5. The last major concern is the similarity  between DEG and imitation learning (IL). Although the paper claims DEG as a RL-based approach, its core mechanism are similar to IL. Specifically, it fine-tunes a pre-trained video generation model on expert data to distill knowledge about robotic manipulation, then creates a supervised signal by comparing the similarity between generated frames and actual policy rollouts. Training a policy to mimic expert behavior using high-quality demonstration data is the fundamental characteristic of IL. Hence, like IL, DEG heavily depends on the availability of high-quality expert data. It would be helpful if authors provide more in-depth analysis on how the quantity and quality of expert data affect DEG, or narrow the scope of the paper to focus on reducing the required expert data instead of claiming DEG as a RL-based approach.

***Minor concerns:***

1. Missing details about important experiment setup, including the RL algorithm used in the simulation experiments, and how many trials are used to compute the success rates, how the expert videos are acquired.
2. The logic of last two sentences of the first paragraph ("To solve this problem, ... in various scenarios") in Section 1 is a bit jumping. How do you conclude that the recent research progress is driven by well-designed rewards?
3. Providing additional statistics to support the two insights in Section 3.1 ("We find that without finetuning, it already possesses the ability ... relevant to the target task") and Section 3.2 ("However, the generated videos inevitably ... their semantically identical real observations") will make the design rationale of DEG more convincing.
4. Typos: Section 2.3 L138 "GEG" -> "DEG"; Section 2.1 L104 ''reward engineering" -> ``reward engineering''.

[a] A vision-language-action-critic model for robotic real-world reinforcement learning.

[b] VLA-RL: Towards Masterful and General Robotic Manipulation with Scalable Reinforcement Learning

[c] Vla-rft: Vision-language-action reinforcement fine-tuning with verified rewards in world simulators

[d] NORA-1.5: A Vision-Language-Action Model Trained using World Modeland Action-based Preference Rewards

**Overall Recommendation:**

Overall, the design of DEG is comprehensive and effective. However, there are several significant weaknesses regarding the current version of this paper. Addressing these concerns would likely require substantial revisions and is not likely to be done during the short period of rebuttal. Hence, my preliminary evaluation leans toward weak rejection.

---

> ### Author Rebuttal · Authors · 2026-03-31
>
> Thanks for your affirmation and constructive reviews! Based on your reviews, we further conduct several additional experiments and improve the paper.  We hope these revisions meet your expectations and that you will consider raising your score.
>
> An anonymous link for additional results: https://anonymous.4open.science/r/DEG_rebuttal-E070
>
> ---
>
> To con1:
>
> We appreciate your reminder! We add corresponding labels to Fig.2 to more clearly indicate which parts of the trajectories are incentivized by two rewards (**Sec.1 via the link**). In addition, we summarize the quantitative comparison using a table (**Sec.11 via the link**).
>
> To con2:
>
> As you pointed out, the number of videos used by DEG is positively correlated with the task number. This is consistent with the baselines. Furthermore, the number of videos used by DEG (3 for most tasks) is much smaller than that of the baselines (60 for TeViR, 20 for Diffusion Reward, and 50 or VIPER).
>
> In addition, we agree that the increase in the number of videos may cause scalability issues.  Follow your suggestion, we conduct additional experiments to show the ability of DEG when faced with multiple task (**Sec.12 via the link**) and less videos (**Sec.8 via the link**), which shows its potential for scalability. Moreover, with the development of foundation models, issues such as data requirements and generalization can be further alleviated. DEG, which can be combined with any generation model, will also benefit from this and achieve better scalability.
>
> Regarding hyperparameters, we agree that this is a limitation of DEG, as they still need to be determined via manual tuning. We further include this in the limitation discussion, and discuss the automatic adjustment of these parameters during training as future work.
>
> To con3:
>
> Thank you for pointing out these important related works. We supplement all your mentioned works and discuss the difference. Details can be founded in **Sec.13 via the link**.
>
> To con4:
>
> Based on your suggestions, we further conduct and supplement several additional experiments.
>
> First, we add large model-based baselines into comparison (**Sec.14 via the link**). DEG shows superior performance upon the supplemented baselines.
>
> Second, we further reduce the number of expert videos in DEG (**Sec.8 via the link**). DEG still enables effective policy learning even with 1 video.
>
> Third, we provide coefficient sensitivity experiments (**Sec.4 via the link**).
>
> Finally, we conduct multitask pre-training (**Sec.12 via the link**). DEG can still work well with a multitask RL guide.
>
> To con5:
>
> We understand your concern. We would like to clarify that DEG is indeed an IL method, yet this does not conflict with that it's a reward design approach for RL. As established in prior work[1], IL can be implemented either via BC, or by inferring rewards and then performing RL. Accordingly, video-based reward design methods such as DEG, diffusion reward, and TeViR can all be regarded as IL approaches.
> Following your suggestions, we first clarify the relationship between IL and DEG in the related work: 'DEG successfully applies large video generation models to RL-based imitation learning.' Then, we provide the experiments of further reducing the expert videos (**Sec.8 via the link**), demonstrating the robustness of DEG when faced with more limited data.
>
> To minor1,q1,q2,q3:
>
> As shown in line326 (page6), DrQv2 is the backbone. DEG results are over at least 3 trails. The expert videos are collected by the official expert provided by Metaworld.
>
> To minor2:
>
> The logic is: these methods aim to improve RL efficiency through different ways, achieving considerable performance under default expert rewards. Thus, they are driven by well-designed rewards.
> We will revise this sentence to eliminate ambiguity: 'These methods effectively improve the RL sample efficiency under the supervision of expert rewards.'
>
> To minor3:
>
> For non-tuned larget model performance, please see **Sec.15 via the link**.
> For more analysis between DEG encoder and pre-trained encoder, please see **Sec.2 via the link**.
>
> To minor4:
>
> Thanks for your careful reading! We have revised them!
>
> To q4:
>
> In one trial, DEG is tested 10 episodes every 10k steps, following DrQv2.
>
> To q5:
>
> Fine-tuning the Wan2.1-I2V-14B model on 3 expert videos for one epoch takes roughly 15+ minutes on one A800 GPU. Since DEG uses 5 epochs, the total fine-tuning time is about 1.5 GPU hours. In RL, since the guidance video is determined only by the initial state of the episode, DEG can reuse these generated guidance videos. For MetaWorld experiments, training naive DrQv2 for 500k steps takes approximately 3.3 hours, and DEG requires a total of 6.4 hours. In comparison, RoboDopamine and Roboreward, which also rely on large models, consume 9.2 hours and 8.7 hours. DEG enables better performance with less time.
>
> [1] A Survey of Imitation Learning: Algorithms, Recent Developments, and Challenges

---

> > ### Author Rebuttal · Reviewer_5RPe · 2026-04-01
> >
> > Thank you for your efforts in preparing the rebuttal. I am impressed by the substantial work the authors have done during this phase. Several of my concerns have been addressed, and I would like to offer a few additional comments on the remaining major issues.
> >
> > 1. **Presentation improvements.**  I appreciate the updates to Figure 2 and the improved presentation of experimental results. While the Figure 2 has been revised, the changes appear relatively minor and still do not fully convey the core ideas behind DEG (e.g., how the two reward components are computed). Anyway, this does not constitute a big problem any more.
> >
> > 2. **Scalability of DEG.**  One of my major concerns is the scalability of DEG, given its reliance on expert videos for each downstream task. The authors demonstrate that DEG can achieve strong performance even with just one expert video per task, which partially alleviates this concern. However, the required number of expert videos still scales linearly with the number of tasks. Moreover, like other video-model-based reward design methods, DEG essentially operates within the imitation learning (IL) paradigm and may therefore be susceptible to distributional shift. While I personally remain skeptical of the philosophy of these approaches (especially many of them claim themselves as “RL-based” but functionally align more closely with IL), this is a common problem across the category rather than a specific problem to DEG. Every  paper has its audience. Given that the authors have already shown good one-shot performance, this is already a meaningful contribution.
> >
> >    A second scalability concern relates to hyperparameter tuning. I agree with the authors’ decision to explicitly acknowledge this as a limitation. Doing so helps readers better understand the practical boundaries of DEG’s applicability and, in my view, enhances the paper’s credibility rather than weakening its contributions.
> >
> > 3. **Related work discussion**.  Thank you for the additional context provided. However, I disagree on the claimed advantages over reward-model or world-model based methods. The authors state: *“DEG differs from prior work in that it does not rely on either learned reward models or action-conditioned world models, but instead formulates reward densification as a contrastive trajectory matching problem, which provides a lightweight yet effective alternative.”*  In my view, this distinction is overstated. Both DEG and these methods train task-specific models—either reward models, world models, or video generation models—for reward densification. Thus, the so-called advantage is not quite proper.
> >
> > 4. **Clarifying the relationship between IL and RL**.  I disagree with the statement intended to put in the paper that *“DEG successfully applies large video generation models to RL-based imitation learning.”* To the best of my knowledge, “RL-based imitation learning” is not a standard or well-defined concept. The authors may be conflating inverse reinforcement learning (IRL), imitation learning (IL), and reinforcement learning (RL). I recommend to accurately reflect DEG’s connection to IL, as outlined in my original Major Concern #5, and claim it as a limitation.
> >
> > 5. **Additional experiments and baselines**.  I appreciate the inclusion of new experimental results. However, I have several follow-up questions:
> >
> >    - How were the **RoboReward** and **RoboDopamine** baselines implemented? Their reported performance appears significantly weaker than what was demonstrated in the original papers. Were there unreported implementation details that might explain this discrepancy?
> >    - Why was **VLA-RFT** not included? I believe this is the most similar and relevant work to DEG, which are highly recommended implemented as baseline. While VLA-RFT does require action-labeled trajectories for world model training, Meta-World datasets do provide such demonstrations, making implementation feasible.
> >    - Finally, why do many baselines, including newly added ones, achieve extremely low or even zero success rates? Given that the test tasks are not quite challenging, such poor performance raises concerns about the experimental setup. For example, is the experimental setup being defined too narrow or inadvertently tailored to benefit DEG?

---

> > > ### Author Response · Authors · 2026-04-05
> > >
> > > We appreciate your detailed feedback! We are delighted that our response has addressed your major concerns on paper presentations (**C1**) and method scalability (**C2**). We hope our responses can further address your remaining concerns and you would consider recommending acceptance of our paper!
> > >
> > > An anonymous link for additional results: https://anonymous.4open.science/r/DEG_rebuttal-E070
> > >
> > > ---
> > >
> > > >To C3:
> > >
> > > Thank you for your feedback. What we intended to convey is that, unlike learned reward models or action-conditioned world models, we don't heavily modify the pre-trained model. Instead, we only use a few video data and the lightweight LoRA fine-tuning to adapt the model to the target task, fully exploiting and leveraging the inherent capabilities of the pre-trained video generation model itself.
> > >
> > > Please see **Sec.16 via the link** for more detailed discussion. We believe this represents the main difference between DEG and others in the utilization of foundation models, and we have added the detailed discussion to the related work section based on your feedback.
> > >
> > >
> > > >To C4:
> > >
> > > We apologize for the confusion caused by inappropriate description. Generally accepted, IL refers to reproducing certain behavior from expert demonstrations. It may be done via BC from action-labeled trajectories, or by extracting rewards and then training a policy using RL. As a reward design method, DEG achieves video imitation learning under the RL paradigm. This results in significantly less distributional shift compared to standard BC-based methods, which is attributed to DEG's interactive trial-and-error process. DEG can explore states outside expert video distribution during training, assign relatively low rewards to them, and learn how to handle them via RL—something that BC-based imitation cannot achieve. In addition, unlike BC methods relying on expert data containing action annotations, DEG uses only videos to derive rewards and does not require actions, which is the key for DEG to improve expert data efficiency with video generation models.
> > >
> > > Based on your comments, we have removed the inappropriate statement (“RL-based imitation”) and added the above discussion to the related work for IL-DEG relationship. In addition, we add the limitations based on your feedback:
> > >
> > > “Although DEG effectively reduces the dependence on the quantity of expert data, it still imposes requirements on the video quality to enable effective adaptation of video generators.”
> > >
> > > >To C5.1：
> > >
> > > As these methods don't report results on Metaworld, we reproduce them by ourselves. We strictly follow their official tutorials for reproduction, using both official models and fine-tuning code. For RoboReward, we directly plugg its official 8B model into the DrQv2 framework. For RoboDopamine, we annotate the fine-tuning videos (same as those used in DEG) following its official instructions, fine-tune its official 8B model, and employ the same RL framework. Their RL backbone and settings are both identical to those of DEG. We provide our analysis regarding their mediocre results, and provide value and score curves per trial to verify the soundness of our training process. See **Sec.17 via the link** for details.
> > >
> > >
> > > >To C5.2:
> > >
> > > We explain why VLA-RFT was not included as the main baseline previously, and following your suggestion, we further add comparative experiments between DEG and VLA-RFT in online RL settings, please refer to **Sec.18 via the link**. Compared with VLA-RFT, the reward designed by DEG is more suitable for low-level control learning from scratch, enabling better policy performance.
> > >
> > > >To C5.3:
> > >
> > > We understand your concern, but we confirm that the experimental settings are not biased toward DEG. We employ the standard DrQv2 framework for all methods, and use the standard MetaWorld setup, following our baselines. We'll release the DEG code after acceptance.
> > >
> > > In fact, directly guiding RL without any environmental feedback in single-camera Metaworld is inherently challenging. Diffusion Reward demonstrates in its paper that it fails consistently on reward-free MetaWorld tasks.TeViR also struggles in the reward-free setting even with three camera views and sufficient videos.
> > >
> > > For the original baselines, the results are fully consistent with those in TeViR or obtained through their official codes on Metaworld. Since DEG uses significantly fewer training steps, the plots are stretched horizontally, which may lead to visual differences.
> > >
> > > For the newly added baselines, we follow their official instructions and plug their official models into the same backbone as DEG. Due to mismatched data distribution and unseen domains, they perform mediocrely on Metaworld from-scratch RL (more details in Response to C5.1 and **Sec.17 via the link**).
> > >
> > > ---
> > >
> > > *Thank you again for your constructive comments and efforts in reviewing our paper! We hope these answers can meet your expectations, and we would be most grateful if you would consider raising your score. Thanks!*

---

### Official Review · Reviewer_hqAw · 2026-03-13

**Soundness:** 3
**Presentation:** 3
**Significance:** 2
**Originality:** 3
**Overall Recommendation:** 4
**Confidence:** 2

**Summary:**

The paper introduces an approach for learning dense rewards from expert videos. The paper uses large video generation models and few expert demonstrations (in the form of videos). More specifically, the paper proposes combining a coarse-level reward, which broadly incentivizes imitating the generated guidance, and a fine-level reward, which incentivizes more fine-grained matching of expert frames. They evaluate on 12 Metaworld manipulation tasks and show their method, DEG, achieves the most interaction-sample efficiency of the algorithms considered, despite using the least amount of expert demonstrations.

**Compliance With Llm Reviewing Policy:**

Affirmed.

**Final Justification:**

Following the rebuttal, the authors have largely addressed my comments, and I'll maintain my positive score. However, due to lack of familiarity with related work, I have lowered my confidence.

**Key Questions For Authors:**

1. Can you leverage the same pretrained video generation model (Wan2.1) in the baselines (e.g. TeViR)?
2. How would different RL algorithms affect the empirical results and the necessity of dense rewards?
3. Suppose that the learner and expert have the exact same starting state. Why would the policy not simply match that initial state perfectly, rendering the fine reward effectively useless? (Equations 7-8)
4. How would leveraging pretrained base policies, which are equipped with exploratory behavior, affect the need for dense rewards?

**Limitations:**

yes

**Strengths And Weaknesses:**

1. At a high-level, DEG incentivizes the learned policy follow similar behavior to the expert demonstrator. However, more specifically, DEG may suffer from two key issues: performance on out-of-distribution (OOD) tasks and performance in misspecified settings.
2. In the first case, if the initial starting state is distinct from the expert demonstrations, it is unclear whether DEG will perform well, and it may suffer from compounding errors. The paper does not provide experiments on OOD tasks (e.g. Libero-OOD).
3. The second setting is misspecification – settings where the learner (i.e. learned policy) cannot perfectly imitate the expert demonstrator. While expert demonstrations may be performed on the same robot through teleoperation and thus avoid misspecification in the action space, there may be misspecification in state space due to perception differences or partial observability.
4. DEG’s method and experimental results are focused on the “RL from scratch” setting. The interaction-sample efficiency and the expert demonstration-sample efficiency of DEG is impressive. However, as acknowledged by the paper, the field of robotics and RL are increasingly focused on leveraging larger generative models. In practice, RL from scratch is rarely performed – instead, RL tends to be initialized with a pretrained policy. In such settings, dense rewards are far less crucial to learning, as the pretrained policy is equipped with some exploratory behavior. However, discussion about how their method works with pretrained policies and experimental evaluations in such settings are entirely missing from the paper.
5. The paper motivates the need for dense rewards as helping to address the exploration problem in RL. However, exploration is also a matter of the RL algorithm employed, as each RL algorithm has methods for inducing exploration, but the paper only uses a single RL algorithm per environment and provides limited discussion about how the choice of RL algorithm could affect results (e.g. the benefits/drawbacks of different exploration mechanisms). Further analysis into the affect of the RL algorithm would substantially strengthen the paper.
6. I appreciate that the authors acknowledge the limitation of video generation speed. However, a runtime comparison to quantitatively evaluate the cost of video generation is needed.
7. The paper provides the values of $\alpha$ and $\beta$ used, which correspond to the strength of the coarse and fine reward, respectively, but it does not discuss how these hyperparameters were chosen. There is an ablation that omits each component, but does not show how sensitive the algorithm is to changing values of each.

---

> ### Author Rebuttal · Authors · 2026-03-31
>
> We really appreciate your careful reading and positive feedback on our paper! Based on your reviews, we further improve the quality of this paper.  We'll answer all your comments below.
>
> An anonymous link for additional results: https://anonymous.4open.science/r/DEG_rebuttal-E070
>
> ---
>
> To c1&c2:
>
> In Metaworld, 3-5 expert videos can't cover all initial states. To this end, performance on Metaworld may already show that DEG can generalize to ood initial states. This is attributed to the pre-trained large model. Based on your feedback, we provide generated videos with unseen initial states (**Sec.3 via the link**), which demonstrates the ood ability.
> Since DEG is designed to provide efficient rewards for learning low-level control policies, Metaworld, which focuses purely on low-level robot policy, may serve as a more suitable benchmark than language-based LIBERO, which follows advanced works[1,2].
>
> To c3:
>
> We agree with your comment! Misspecifications, such as object occlusion caused by single-view camera, can indeed affect the performance. As a common issue, recent works[2,3] adopt multiple cameras to mitigate this problem. In contrast, DEG aims to explore the upper bound under single-camera setting, a more challenging and general setup.
>
> To c4:
>
> As you point out, we mainly adopt from-scratch training. This is primarily because DEG is designed as a pure reward-shaping method for low-level control learning. We aim to highlight the quality of the reward itself and avoid comparisons between different rewards being affected by other components, thus employing the standard experimental setup, which follows recent works[1,2].
> Moreover, DEG is not restricted to from-scratch learning. In real-world tasks, the backbone HIL-SERL employs expert trajectories for pretaining before RL, which can be regarded as a non-from-scratch RL paradigm. In this setting, the dense reward provided by DEG still yields significant performance improvements.
>
> To c5:
>
> In experiments, we employ a widely adopted backbone for all methods in Metaworld: DrQv2, for fair comparisons[1,2]. Taking your comment into consideration, we further modify the exploration strategy of the RL method to investigate DEG’s performance with a stronger exploration backbone. The results (**Sec.10 via the link**) show that using a backbone with stronger exploration does improve the efficiency of exploring sparse success rewards, but its performance is still far inferior to that with DEG dense rewards.
>
> To c6:
>
> Thanks for your kind comments. Fine-tuning the Wan2.1-I2V-14B model on 3 expert videos for 1 epoch takes roughly 15+ minutes on one A800 GPU. Since DEG uses 5 epochs, the total fine-tuning time is about 1.5 GPU hours. In fact, the model already achieves strong domain adaptation after just 1 epoch.
> In RL, since the guidance video is determined only by the initial state of the episode, DEG can reuse these generated guidance videos, which significantly improves overall training efficiency. For MetaWorld experiments, training naive DrQv2 for 500k steps takes approximately 3.3 hours, and DEG requires a total of 6.4 hours. In comparison, RoboDopamine and Roboreward, which also relies on large models, consumes 9.2 hours and 8.7 hours. DEG enables better performance with less time (results in **Sec.14 via the link**).
>
> To c7:
>
> Regarding the coefficients among different rewards, our goal was to scale them to a similar order of magnitude (range from 10 to 100). This setup already yielded satisfactory results. Following your suggestion, we further conduct a sensitivity analysis on these parameters (**Sec.4 via the link**).
>
> To q1:
>
> We agree that this is a meaningful comparative experiment. However, since TeViR doesn't provide the implementation, it would be challenging for us to set up the codebase and ensure an appropriate comparison within the short period of the rebuttal. For better comparison, we further compare DEG with advanced VLM-based rewards (**Sec. 14 via the link**).
>
> To q2:
>
> As shown in response to c5, Using an RL backbone with stronger exploration can indeed improve sample efficiency under sparse rewards, but its performance remains far inferior to using DEG's dense rewards.
>
> To q3:
>
> As you point out, DEG triggers the fine-grained reward at initial stage. Since the index of the matched frame is 0, it does not affect the matching between subsequent frames with larger indices (1,2,3….) and the agent's observations, allowing the fine-grained reward to remain effective throughout the later learning process.
>
> To q4:
>
> As shown in response to c4, HIL-SERL is a backbone with a pre-trained policy before RL. The experiments (**Sec.10 via link**) also demonstrate DEG's effectiveness with the more exploratory backbone.
>
> [1] Diffusion reward: Learning rewards via conditional video diffusion
>
> [2] TeViR: Text-to-Video Reward With Diffusion Models for Efficient Reinforcement Learning
>
> [3] Robo-Dopamine: General Process Reward Modeling for High-Precision Robotic Manipulation

---

> > ### Author Rebuttal · Reviewer_hqAw · 2026-04-03
> >
> > Thanks for the detailed responses. I'll maintain my score.

---

> > > ### Author Response · Authors · 2026-04-05
> > >
> > > Thank you for your positive feedback and recognition!
> > >
> > > We are very pleased that our responses have addressed your concerns and that you recommend accepting our paper. We sincerely appreciate your valuable comments and dedicated efforts for reviewing our paper!

---

### Official Review · Reviewer_rdBx · 2026-03-13

**Soundness:** 3
**Presentation:** 3
**Significance:** 2
**Originality:** 3
**Overall Recommendation:** 4
**Confidence:** 4

**Summary:**

This paper proposes DEG, a reward-design pipeline for embodied RL that LoRA-finetunes a large image-to-video generator using expert clips to produce an episode-conditioned “guidance video” from the initial observation, then trains a domain-specific encoder with contrastive self-supervision to align generated frames with real observations in latent space, and uses a dual-granularity dense reward combining a coarse sequential progress term and a fine non-sequential matching term. The core claim is that this yields sample-efficient RL without human annotations, and can either stand alone or be combined with sparse success rewards. Empirically, the paper reports gains on 12 reward-free tasks, competitive performance versus expert dense rewards on 16 tasks (with sparse success), and improvements on two Franka real-world tasks.

**Compliance With Llm Reviewing Policy:**

Affirmed.

**Final Justification:**

The rebuttal has addressed my concerns. I'll maintain my positive score.

**Key Questions For Authors:**

- Coarse reward enforces sequential progress, fine reward allows jumps, could these two incentives conflict and create non-stationarity or oscillatory learning?
- Have you analyzed whether your reward can be expressed as (or approximates) potential-based shaping so as not to alter the task optimum?
- You argue a loose threshold avoids confinement to one trajectory. But the reward still uses a single generated video. How do you prevent bias toward that specific imagined path, especially in tasks with multiple valid strategies?

**Limitations:**

yes

**Strengths And Weaknesses:**

pros

- The paper focuses on an important topic of robotics.
- The paper is well written and easy to follow.

cons

- The fine-grained reward returns the index of the matched frame (up to video length), which is a strange, discontinuous shaping signal with a large, time-varying magnitude.
- The Appendix prompts are task-specific, and essentially encode a lot of human knowledge about geometry, colors, object names, physics constraints, and even environment quirks (eg, rebound behavior). This is not “annotation-free” in any meaningful sense, it’s closer to prompt engineering as supervision plus a curated sim aesthetic.
- If you already have 3–5 expert videos, obvious comparisons are: BC, BC+RL fine-tuning, advantage-weighted imitation, offline RL, or even “nearest-neighbor video reward” using the real expert clips without a generator. Without these, it’s unclear whether the win comes from “episodic generation” or just from “any demonstration-shaped reward.”

---

> ### Author Rebuttal · Authors · 2026-03-31
>
> Thanks for your careful reading and positive feedback! Based on your reviews, we further improve our paper. We hope the following answers can address your concern!
>
> An anonymous link for additional experiments: https://anonymous.4open.science/r/DEG_rebuttal-E070
>
> ---
>
> To cons1:
>
> We understand your concern. While fine-grained rewards are discontinuous, this is not a critical issue, as coarse-grained rewards provide a continuous and smooth signal. The fine-grained reward additionally rewards precise matches, based on the policy shaped by the smooth coarse rewards.
> Furthermore, fine-grained rewards are based on frame indexes. The main motivation is that when the agent matches later expert frames, it is typically closer to completing the task than when matching earlier frames, and thus should receive higher rewards.
>
> To cons2:
>
> We agree with your comment! Although describing a video is much easier than annotating much data, it still involves manual knowledge. Following your suggestion, we revise the relevant description to “with as little expert data and manual annotation as possible” to better match the method.
> We provided detailed prompts to maximize generation quality and verify the idea of using a large video model as an RL guide. Based on your feedback, we further conduct fine-tuning with simple prompts (**Sec.5 via the link**). DEG still enables effective RL with a brief prompt ‘open the drawer’.
>
> To cons3:
>
> We agree with your point! Due to the absence of action and reward labels, directly performing BC or offline RL on videos can be challenging. Following your suggestions, we test the nearest-neighbor video reward (**Sec.9 via the link**), showing the importance of episodic guidance.
>
> To q1:
>
> Coarse-grained and fine-grained rewards don't conflict as they share the same optimal objective of achieving target tasks. They differ in their roles, which are inherently complementary. In Appendix B (page 18 in the paper), the ablation shows that they both contribute to good performance. Visualizations further support their motivation: The coarse reward uses continuous signals to encourage sequential exploration, enabling efficient imitation of the arm’s motion intent. It doesn't explicitly account for whether the interaction with the drawer is successful. The fine-grained reward further exerts its capability based on the high-quality exploration brought by the coarse reward. It further rewards successful interaction and subsequent drawer-pulling frames, which refines the policy from rough to fine manipulation.
>
> To q2:
>
> Thank you for this insightful question. We agree that our dual-granularity contrastive reward is closely related to potential-based reward shaping.
> For the coarse-grained reward, the cosine similarity $S(t,i)$ can be naturally interpreted as a task-relevant potential function $\Phi(s)$. The resulting reward takes the form $\Phi(s_t) - \Phi(s_{t-1})$, which is consistent with standard potential-based shaping and thus does not alter the optimal policy.
> For the fine-grained reward, however, the reward does not strictly follow the potential-based formulation. Specifically, it depends on the maximum similarity observed along the trajectory and includes a threshold-based mechanism to encourage meaningful progress without repeatedly rewarding the same state. As a result, this component introduces a mild form of history dependence and cannot be fully expressed as a pure potential difference.
> Nevertheless, we emphasize that the fine-grained reward is still grounded in the same similarity metric $S(t,i)$ and is aligned with the task objective (i.e., reaching the reference goal). Empirically, we observe that it mainly serves as an auxiliary signal to improve exploration efficiency rather than altering the underlying task optimum.
> Therefore, while the coarse-grained reward strictly satisfies potential-based shaping, the overall dual-granularity reward can be viewed as a potential-based shaping term augmented with an auxiliary progress bonus, which may slightly deviate from the theoretical guarantees but is practically effective.
>
> to q3:
>
> In fact, we don't avoid the agent’s reliance on a single trajectory. Even if multiple good trajectories exist, the agent can still complete the task if it can perfectly imitate the generated trajectory. However, during training, we observed that enforcing perfect matching to the target trajectory is hard, as it makes the agent overly preoccupied with non-critical frames that may not affect success, thereby harming the sample efficiency. We therefore revised our objective from ‘strict trajectory matching’ to ‘task completion with permissible deviations’. To this end, we propose dual-grained rewards that tolerate trajectories deviating from expert guidance yet still achieving the task. By relaxing unnecessarily stringent constraints on the agent, DEG achieves improved efficiency and performance, while it does not reject precise imitation of the target trajectory in essence.

---

> > ### Author Rebuttal · Reviewer_rdBx · 2026-04-03
> >
> > Thanks for the detailed response. I'll maintain my positive score.

---

> > > ### Author Response · Authors · 2026-04-05
> > >
> > > Thank you for your recognition of our response!
> > >
> > > We are delighted that our answers have addressed your concerns and that you recommend accepting our paper. We sincerely appreciate your constructive comments and dedicated efforts during the review and rebuttal phases.

---

### Official Review · Reviewer_M57f · 2026-03-15

**Soundness:** 2
**Presentation:** 1
**Significance:** 2
**Originality:** 3
**Overall Recommendation:** 4
**Confidence:** 4

**Summary:**

The paper proposes a method for designing dense reward functions using a minimal amount of expert video demonstrations, without reliance on manual annotations. It introduces a dual-granularity reward that combines fine-grained expert behavior–matching rewards with coarse-grained exploration rewards, effectively balancing exploitation and exploration. The proposed approach requires only a small set of expert videos (3–5 clips) to fine-tune large pre-trained video models. The fine-tuned model then serves as a guide for generating additional expert-like video demonstrations. The authors employ Wan2.1-I2V-14B as the video backbone and use Low-Rank Adaptation (LoRA) for efficient fine-tuning. Contrastive learning is applied to align encoder representations between generated samples and expert demonstrations.

**Compliance With Llm Reviewing Policy:**

Affirmed.

**Final Justification:**

The authors responded and addressed my concerns and questions clearly. The scores is updated to positive, 4.

**Key Questions For Authors:**

* Please provide the experimental details including prompt sensitivity, hyperparameters, the number of seeds, and etc. This is essential for reproducibility and practical deployment.
* Do you have any reason to have different task sets in Figure 3 and Figure 4? This gives an impression that the authors might cherry pick the results. TeViR (not published) results do not match original paper? Consistently presenting and comparing results for both DEG and DEG+ are recommended. Any discussions regarding the impact of increasing or decreasing the number of expert video clips?
* What was the impact of $\alpha, \beta, \theta$ selection? Any computational codes discussions?

**Limitations:**

The authors briefly mention the limitation of slow video generation in the DEG module in the conclusion. While this is understandable given the scope of the work, a more detailed discussion of generation speed and the potential practical implications of this limitation would be beneficial.

**Strengths And Weaknesses:**

### Strengths

* sample efficiency, requiring minimal expert videos and improved performance over baselines.
* Extensive experiments, ablation studies, and visualizations

### Weaknesses

* The writing is at times difficult to follow and lacks sufficient justification and reasoning for several design choices.
* Key experimental details are missing, and the paper lacks clear and informative visualizations illustrating the training dynamics of the DEG encoder.
* There is no discussion or empirical investigation of hyperparameters choices and prompt design/sensitivity.
* The approach requires extensive fine-tuning with video generation makes the method computationally expensive.

---

> ### Author Rebuttal · Authors · 2026-03-31
>
> Thank you for reading our paper carefully and affirming our paper! Following your constructive comments, we have revised our expressions and supplemented additional experiments to further improve the paper’s quality. We hope these revisions meet your expectations and that you will consider raising your score. Thank you!
>
> An anonymous link for additional results: https://anonymous.4open.science/r/DEG_rebuttal-E070
>
> ---
> To w1：
>
> We apologize that the current version did not clearly convey the design motivation of the DEG module, making following improvement:
> First, regarding the reward module, we further update the main figure (**Sec.1 via the link**) and move the visualization of the policy intentions corresponding to different sub-rewards from the Appendix to the main text (Now **Fig.9, page 18**), to clearly illustrate the motivation and effect of dual-granularity reward: The coarse-grained reward enables efficient learning of the robotic motion intent but lacks attention to fine-grained precise interactions. In contrast, the fine-grained reward focuses sufficiently on interactions but fails to capture the overall motion intent of the arm. The two components complement each other’s strengths and weaknesses, and their combination enables efficient and fine-grained policy learning.
> Second, for the reasons of DEG encoder, we add a more intuitive comparison of latent space capabilities between DEG encoder and DINOv3, see response to w2 .
> Finally, we have supplemented results of the fine-tuned video generation model on both seen and unseen initial states (**Sec.3 via the link**) to show why DEG chooses video generation model.
>
> To w2:
>
> The hyper-parameters of DEG are provided in the Table 1 (Appendix A.3, page 14). The results of DEG are over 3 different trials.
> Based on your feedback, we provide more detailed visualizations of the DEG encoder and the DINOv3 encoder (**Sec.2 via the link**), which more intuitively demonstrate the advantages of the DEG encoder: it exhibits a stronger ability to align semantic distances with latent space distances, as well as better robustness to generated noise.
>
> To w3:
>
> Following your suggestion, we further provide sensitivity experiments on hyper-parameters (**Sec.4 via the link**).
> In addition, we also conduct prompt sensitivity experiments (**Sec.5 via the link**), demonstrating that DEG can also work well with simple prompts.
>
> To w4:
>
> We understand your concerns, while on most tasks, DEG only uses 3 expert videos for learning. This number is significantly lower than the data used by baseline methods (TeViR uses 60 three-view videos per task, Diffusion Reward uses 20 videos per task, while VIPER uses 50 videos per task.) Furthermore, we would like to clarify that DEG presents the idea of using large video generation models as an episodic guide for RL. It can be combined with any existing or future foundation models. We believe that the video requirement of DEG can be further reduced as foundation models continue to advance.
>
> To q1:
>
> The hyper-parameters of DEG are provided in the Table 1 (**Appendix A.3, page 14**). The results of DEG are over 3 trials. The sensitivity experiments on prompts and sub-reward coefficients are provided in the **Sec.4 & Sec.5 via the link**. We'll release the code after acceptance.
>
> To q2:
>
> For task sets: Learning without sparse success rewards poses significant challenges in Metaworld. For instance, Diffusion Reward demonstrates that it fails to guide effective RL alone in its paper, and TeViR only selects 8 out of 11 tasks as reward-free domains. We follow TeViR, adopting all 8 reward-free environments. For some harder tasks, all methods perform not well without sparse rewards, so we don’t include it in the figure and appropriately reduced the number of tasks to improve the visual layout of the figure (from 4 rows to 3 rows, totaling 12 tasks). While DEG still achieves better performance on these tasks (**Sec.6 via the link**).
>
> For TeViR: We contacted the authors of TeViR and directly used the original learning curves of their method and the baselines on the reward-free tasks. Since DEG employs significantly fewer training steps, the plots are stretched horizontally, which may lead to visual differences.
>
> For DEG & DEG+: Based on your suggestions, we further add the curves of DEG in DEG+ figure (**Sec.7 via the link**).
>
> For number of videos: To answer your questions, we further conduct additional experiments on the number of video clips (**Sec.8 via the link**). DEG and DEG+ can also work well with less videos, while more videos are better if possible.
>
> To q3:
>
> For the sub-reward coefficients, we roughly scaled them to the same order of magnitude without extensive additional tuning. To answer your question, we further conduct sensitivity analysis on these parameters (**Sec.4 via the link**).

---

> > ### Author Rebuttal · Reviewer_M57f · 2026-04-03
> >
> > Thank you for the detailed responses with the anonymous link.
> >
> > Regarding the W4, I appreciate the reduction of the number of expert videos for learning. However, the authors have not discussed/replied regarding the computational costs for fine-tuning and video generation. This can be related to the scalability comment by Reviewer 5RPe. Please discuss this cost and how it can impact on scalability for downstream tasks and if it is costly for now, how it can be improved.

---

> > > ### Author Response · Authors · 2026-04-05
> > >
> > > Thank you for careful reading and for your recognition of our response! We are pleased that most of your concerns have been addressed. We will address your remaining comments and concerns one by one. We hope our responses in this round will meet your satisfaction and you would consider raising your score. Thank you!
> > >
> > > An anonymous link for additional results: https://anonymous.4open.science/r/DEG_rebuttal-E070
> > >
> > > ---
> > >
> > > >For the computational costs for finetuning and video generation & impact:
> > >
> > > We apologize for forgetting to include the response regarding computational cost in the Response to W4, which results in your concern.
> > >
> > > Fine-tuning the Wan2.1-I2V-14B model on 3 expert videos for 1 epoch takes roughly 15+ minutes on one A800 GPU. Since DEG uses 5 epochs for default, the total fine-tuning time is about 1.5 GPU hours. In fact, we observe that the pre-trained model can already achieve strong domain adaptation after 1 epoch.
> > >
> > > In downstream RL, since the guidance video is determined only by the initial state of the episode, DEG can reuse these generated guidance videos when faced with seen initial states, which significantly improves overall training efficiency. For MetaWorld experiments, training naive DrQv2 for 500k steps takes approximately 3.3 hours, and DEG requires a total of 6.4 hours, which means 3.1 additional GPU hours for video generation and reward computation. In comparison, RoboDopamine and Roboreward, which also rely on large models, consume 9.2 hours and 8.7 hours respectively. DEG enables better performance (results in **Sec.14 via the link**) with less time than these two large model-based approaches.
> > >
> > > We agree that the computational resources and expert data required by DEG increase in proportion to the number of tasks, which affects its scalability to a certain degree. We would like to note that this is a common issue shared by our baseline methods, and DEG effectively alleviates this issue via its significantly lower dependence on data volume and human annotations.
> > >
> > > In summary, while DEG does introduce additional computational overhead, it is still acceptable, especially given its extremely low reliance on data volume and human annotations, as well as its superior performance compared to baseline methods.
> > >
> > > Based on your feedback, we have added the DEG computational resources to the experiments section, and supplemented the discussion of scalability impact to the limitation section.
> > >
> > > >For future improvement:
> > >
> > > DEG is designed to be easily combined with any video generation models. With the rapid development of foundation models, issues such as data requirements and computational cost can be further alleviated, and DEG will also benefit from this, gaining corresponding improvement.
> > >
> > >
> > >
> > > >For scalability comment by Reviewer 5RPe:
> > >
> > > Thanks for your careful reading! Based on the above discussion on computational cost and the additional scalability experiments (including the fewer videos experiments in **Sec.8 via the link** and multi-task experiments in **Sec.12 via the link**), Reviewer 5RPe has stated that their major concerns regarding scalability have been resolved and acknowledged the contributions of our paper.
> > >
> > > ---
> > >
> > > *Thank you again for your constructive comments and efforts in reviewing our paper! We hope our responses can address your remaining concerns, and we would be most grateful if you would consider recommending our paper!*

---

### Decision · Program_Chairs · 2026-04-30

**Decision:**

Reject

**Comment:**

This paper joins the effort of designing suitable reward functions with video demos. In real-world reward design for RL problems, embodied manipulation reward design is on the easy side instead of the difficult side.

This paper investigates the problem of learning rewards from expert demo videos.

The authors presented a reward-design pipeline that fine-tunes a large image-to-video generator using expert clips with LoRA. And a domain-specific encoder with contrastive self-supervision to align generated frames with real observations in latent space. Combining a coarse-level reward and a fine-level reward can balance generation and imitation. The authors evaluated their method on Meworld manipulation tasks with good efficiency.

However, I do not think the current version is strong enough for acceptance. While the rebuttal addressed several concerns, important questions remain about scalability, computational cost, and the overall positioning of the method.